

# NALPS19: Sub-orbital scale climate variability recorded in Northern Alpine speleothems during the last glacial period

Gina E. Moseley[1], Christoph Spötl[1], Susanne Brandstätter[1], Tobias Erhardt[2], Marc Luetscher[1*], R. Lawrence Edwards[3]

[1]Institute of Geology, University of Innsbruck, Innrain 52, 6020 Innsbruck, Austria

[2]Climate and Environmental Physics and Oeschger Center for Climate Change Research, University of Bern, Sidlerstrasse 5, 3012 Bern, Switzerland

[3]School of Earth Sciences, University of Minnesota, John T. Tate Hall, Room 150, 116 Church Street SE, Minneapolis, MN 55455-0149, USA

[*]Present address: ML Swiss Institute for Speleology and Karst Studies (SISKA), CH-2301 La Chaux-de-Fonds, Switzerland

*Correspondence to:* Gina E. Moseley (gina.moseley@uibk.ac.at)

**Abstract.** Sub-orbital-scale climate variability of the last glacial period provides important insights into the rates that the climate can change state, the mechanisms that drive that change, and the leads, lags and synchronicity occurring across different climate zones. Such short-term climate variability has previously been investigated using speleothems from the northern rim of the Alps (NALPS), enabling direct chronological comparisons with highly similar shifts in Greenland ice

cores. In this study, we present NALPS19, which includes a revision of the last glacial NALPS $\delta^{18}$O chronology over the interval 118.3 to 63.7 ka using eleven, newly-available, clean, precisely-dated stalagmites from five caves. Using only the most reliable and precisely dated records, this period is now 90 % complete and is comprised of 15 stalagmites from seven caves. Where speleothems grew synchronously, major transitional events between stadials and interstadials (and vice versa) are all in agreement within uncertainty. Ramp-fitting analysis further reveals good agreement between the NALPS19

speleothem $\delta^{18}$O record, the GICC05$_{modelext}$ NGRIP ice-core $\delta^{18}$O record, and the Asian Monsoon composite speleothem $\delta^{18}$O record. In contrast, NGRIP ice-core $\delta^{18}$O on AICC2012 appears to be considerably too young. We also propose a longer duration for the interval covering Greenland Stadial (GS) 22 to GS-21.2 in line with the Asian monsoon and NGRIP-EDML. Given the near-complete record of $\delta^{18}$O variability during the last glacial period in the northern Alps, we offer preliminary considerations regarding the controls on mean $\delta^{18}$O. We find that as expected, $\delta^{18}$O values became increasingly more

depleted with distance from the oceanic source regions, and increasingly depleted with increasing altitude. Exceptions were found for some high-elevation sites that locally display $\delta^{18}$O values that are too high in comparison to lower-elevation sites, thus indicating a summer bias in the recorded signal. Finally, we propose a new mechanism for the centennial-scale stadial-level depletions in $\delta^{18}$O such as 'pre-cursor' events GS-16.2, GS-17.2, GS-21.2, and GS-23.2, as well as the 'within-interstadial' GS-24.2 event. Our new high-precision chronology shows that each of these $\delta^{18}$O depletions occurred shortly

following rapid rises in sea level associated with increased ice-rafted debris and southward shifts in the Intertropical Convergence Zone, suggesting that influxes of meltwater from moderately-sized ice sheets may have been responsible for the cold reversals causing the AMOC to slow down similar to the Preboreal Oscillation and Older Dryas deglacial events.

## 1 Introduction

Speleothems from the northern rim and central European Alps have provided a number of important, high-resolution,

precisely [230]Th-dated records of both orbital and millennial-scale climate variability during the last glacial and interglacial periods (Spötl and Mangini, 2002; Spötl et al., 2006; Boch et al., 2011; Moseley et al., 2014; Luetscher et al., 2015; Moseley et al., 2015; Häuselmann et al., 2015). The isotopic signature of such records have helped improve fundamental understanding of the effect that changes in atmospheric (Luetscher et al., 2015) and North Atlantic circulation (Moseley et al., 2015) have on European climate, whilst the robust chronologies have provided important information about the

timescales upon which the climate can change in this well-populated region (Boch et al., 2011; Moseley et al., 2014).



Furthermore, the pattern and timing of isotopic excursions in δ[18]O as recorded in the calcite of northern Alpine speleothems during the last glacial cycle have been shown to be synchronous within dating uncertainties with the sawtooth-pattern of changes in the δ[18]O of Greenland ice cores (known as Dansgaard-Oeschger cycles; Johnsen et al., 1992; Dansgaard et al., 1993), thus reflecting the shared North Atlantic moisture source and integrated climate system (Boch et al., 2011). The

sawtooth pattern of δ[18]O is generally interpreted in both Greenland and the northern Alps as being caused by a rapid increase in temperature and humidity leading into a mild climate state (interstadial), followed by a gradual cooling leading into a cold and dry glacial state (stadial). In total, 25 such cycles of rapid warming and gradual cooling are recognised as having occurred during the last glacial period (Dansgaard et al., 1993; NGRIP Project members, 2004), though with higher-resolution data now available, smaller centennial and decadal-scale events are increasingly being recognised (Capron et al.,

2010a; Moseley et al., 2014). This has resulted in a new stratigraphic framework for abrupt climate changes in Greenland, in which shorter-scale events that occur within the 25 main stadials and interstadials are designated "a to e" (Rasmussen et al., 2014). This nomenclature will be used in the remainder of this manuscript.

When considering the timing of the transitions between stadial and interstadial states, the largest offsets between the northern Alps speleothem chronology (NALPS) and Greenland ice-core chronologies (GICC05; Svensson et al., 2008 and

GICC05$_{modelext}$; Wolff et al., 2010) are 767 years in Marine Isotope Stage (MIS) 3 (Moseley et al., 2014) and 1,060 years in MIS 5 (Boch et al., 2011). The former is associated with the warming transition into Greenland Interstadial-16.1c (GI-16.1c), and the latter with the cooling transition into Greenland Stadial-22 (GS-22; Rasmussen et al., 2014). The timing for both of these transitions in the NALPS chronology was constrained from speleothems high in detrital thorium (Boch et al., 2011; Moseley et al., 2014). Since one of the prerequisites for reliable [230]Th dating is that minimal [230]Th is incorporated into the

crystal lattice at the time of deposition (Ivanovich and Harmon, 1992; Dorale et al., 2004), it is reasonable to question the accuracy of the age of these two transitions. In the case of the MIS 3 sample (Moseley et al., 2014), the correction for the initial incorporation of daughter nuclides was well constrained by isochron methods (Ludwig and Titterington, 1994; Dorale et al., 2004), however, in the case of the MIS 5 sample (Boch et al., 2011), the detrital Th was corrected for using an *a priori* assumption that the contaminant phase had the same composition of the silicate bulk earth (Wedepohl, 1995). Furthermore,

the accuracy of the GICC05$_{modelext}$ chronology is questionable in the vicinity GI-22 to 21 (Capron et al., 2010b; Vallelonga et al., 2012). Specifically, the duration of GS-22 appears to be underestimated, probably as a result of an overestimation of the annual layer thickness by the ss09sea06bm ice flow model upon which GICC05$_{modelext}$ is based in the portion of the record older than 60 ka (Wolff et al., 2010; Vallelonga et al., 2012). Vallelonga et al. (2012) thus revised the duration of GS-22 from 2,620 years to 2,894 ± 99 years using annual layer-counting of seasonal cycles in the chemical impurities in the ice.

Given the uncertainties in the chronologies for both the NALPS speleothems and NGRIP ice core during GI-22 to 21, it is thus difficult to determine the reliability and extent of the leads, lags and synchronicity at this time. In addition to the complexities around GS-22, the chronology of events between GI-25 to 23 are also poorly constrained. This is visible when comparing the GICC05$_{modelext}$ chronology (Wolff et al., 2010) with the AICC2012 chronology (Veres et al., 2013), and also when comparing the pattern of the δ[18]O shifts during GS-24 in NALPS and NGRIP (Boch et al., 2011).

Here, we revisit the NALPS speleothem chronology over the interval 63.7 to 118.3 thousand years ago (ka)(Boch et al., 2011) using new samples that are low in detrital thorium and/or have a more pronounced δ[18]O signal, with the aim of improving the chronology such that better informed conclusions about leads/lags and synchronicity in the climate system may be made. The original record was discontinuous, with coverage of 76 % of the 54.6 ka interval. Gaps in the record were present between 111.6 and 110.0, 94.5 and 89.7, 84.7 and 83.0, 77.5 and 76.0, 75.5 and 72.0 ka (Boch et al., 2011). With the

addition of new speleothems, we extend the coverage to 90 %, improve the accuracy and precision of some climate transitions, and designate the revised chronology "NALPS19". Using the ramp-fitting method of Erhardt et al. (2018), we find good agreement in the timing of δ[18]O shifts recorded in NALPS19 (this study), Asian composite monsoon (Kelly et al., 2006; Cheng et al., 2016), and NGRIP on GICC05$_{modelext}$ (Svensson et al., 2008; Wolff et al, 2010). Furthermore, the near-



complete record of δ¹⁸O variability along the northern rim of the Alps during the last glacial period enables a deeper consideration of the main factors controlling the δ¹⁸O of precipitation. We also consider a meltwater-triggered mechanism for the centennial-scale cold reversals of GS-16.2, GS-17.2, GS-21.2, GS-23.2 and GS-24.2 similar to those already recognised for the deglacial period.

### 1.1 Regional Climate

The European Alps, situated between 44 and 48 °N, are a 950 km-long mountain range running ENE- WSW located close to the southern fringe of the European mainland. The highest peaks, reaching over 4,000 m in elevation, are situated in the Western Alps of France and Switzerland, whilst the Eastern Alps, located in Austria, are on average 1,000 m lower. Across the whole of the Alps, the average elevation is c. 2,500 m above sea level (a.s.l.), thus this mountain range forms a major topographic barrier between the North Atlantic and Mediterranean climate zones (Wanner et al., 1997). Today, the Alps are located to the south of the extra-tropical westerlies, which bring precipitation sourced from the North Atlantic to the northern and western flanks, particularly during winter and spring (Wanner et al., 1997; Sodemann and Zubler, 2010). Lagrangian back-trajectory studies have shown that for the period 1995-2002, the North Atlantic-contributed c. 40% of the annual mean moisture to the Alps, whilst the Mediterranean contributed 23%, the Arctic, Nordic and Baltic seas 16%, and the European land masses 21% (Sodemann and Zubler, 2010). Contributions to the northern versus southern side of the Alps, however, displayed considerable seasonal differences. Throughout the year, the North Atlantic contributes more moisture to the northern Alps as compared to the southern Alps, and this is especially pronounced in winter and spring (Sodemann and Zubler, 2010). During summer, Central European land masses are the dominant moisture source across the entire Alps, though the North Atlantic still makes some contribution to the northern flanks, and the Mediterranean to the southern flanks. In autumn, the northern Alps receives comparable quantities of moisture from both the North Atlantic and Mediterranean, whilst the southern Alps are dominated by moisture from the Mediterranean (Sodemann and Zubler, 2010). On longer, multi-decadal timescales, moisture sources and trajectories to the Alps have been shown to be highly variable.

One of the dominant controls is the phase of the North Atlantic Oscillation (NAO), which is especially pronounced in winter (Wanner et al, 1997). During the positive phase, when positive sea-surface temperature and air-pressure anomalies build up in the southwestern North Atlantic, and negative ones in the north, the associated temperature gradient across the western North Atlantic is high. This leads to an intensification of the North Atlantic polar front jet stream, which creates a high pressure zone over the Alps and Mediterranean causing higher temperatures and less precipitation (Wanner et al., 1997). Conversely, during a negative NAO phase the air pressure decreases over the Alps and Mediterranean leading to lower air temperatures and higher precipitation.

## 2 Methods

### 2.1 Cave Sites and Speleothems

Previous NALPS studies include MIS 2 in Luetscher et al., (2015)(though this was not branded as 'NALPS'), MIS 3 in Moseley et al., (2014), and MIS 4 to MIS 5 in Boch et al., (2011). The MIS 4/ MIS 5 chronology (which is the part revised here), was constructed from seven speleothem samples from four cave sites including; St. Beatus caves, Große Baschg cave (Baschg cave for short), Klaus-Cramer cave and Schneckenloch (Boch et al., 2011). In this study, two additional samples from Baschg cave and one from Schneckenloch cave were analysed, plus one sample from Hölloch im Mahdtal (Hölloch cave for short), one from Grete-Ruth-Shaft, and six from Gassel Tropfsteinhöhle (Gassel cave for short). All cave sites are located on the northern rim of the European Alps (Fig. 1) and have small catchments of less than a few km². The distance between the most westerly and easterly caves is c. 475 km.



Moving from west to east and considering the new samples analysed in this study, Baschg cave (47.2501 N, 9.6667 E) is a 300 m-long cave with a single-known entrance located at 785 m a.s.l.. The cave air temperature is c. 10 °C and the mean annual precipitation 1,360 mm (recorded between 1981-2010 at the Feldkirch meteorological station located c. 5 km WNW from the cave at 438 m a.s.l. (ZAMG, 2018)). The nearest GNIP station is located 20 km SW at Sevelen where $\delta^{18}$O ranges

between -6.3 in July to -15.8 ‰ in November (IAEA, 2018). Samples from Baschg cave analysed in this study include BA-5 and BA-7 (Supplementary Information (SI) Fig. 1). Both are honey-brown coloured stalagmites c. 70 mm and 200 mm in length respectively. BA-5 and BA-7 were collected from the rear of the cave, c.180 m from the entrance. BA-5 was found buried in loam above the streamway, and BA-7 was found broken above the streamway. The base of BA-7 remains in situ. Schneckenloch (47.3745 N, 10.0680 E), Klaus-Cramer (47.3559 N, 10.1064 E), and Hölloch (47.3779 N, 10.1505 E) caves

are located within 10 km of one another. Schneckenloch is a 3.5 km-long cave with an entrance at an elevation of 1,285 m a.s.l. (Klampfer et al., 2017). The cave air temperature is c. 6 °C and the mean annual precipitation 2,073 mm (recorded at the Schoppernau meteorological station located c.7 km SSW from the cave at 839 m a.s.l. (ZAMG, 2018)). The nearest GNIP station is located 50 km WNW at St. Gallen where $\delta^{18}$O ranges between -6.9 in July to -15.0 ‰ in February (IAEA, 2018). In Schneckenloch, speleothem deposition takes place today in the form of stalactites and active stalagmite and

flowstone deposition is rare. For this study, a honey-brown coloured stalagmite (SCH-6), c. 235 mm in length was recovered from the end of a small, well-decorated side passage, 350 m from the entrance.

Hölloch is a 10.9 km-long cave located c. 10 km to the east of Schneckenloch. It has two known entrances at elevations of 1,240 m and 1,438 m a.s.l. (Wolf, 2006). The cave air temperature in the lower part is 5.6 ± 0.2 °C (Spötl et al., 2011). Mean annual precipitation is the same as for Schneckenloch. Hölloch is located c. 57 km ESE to the St. Gallen GNIP station, and

70 km WSW to the Garmisch-Partenkirchen GNIP station. At Garmisch-Partenkirchen $\delta^{18}$O ranges between -6.7 in July to -14.7 ‰ in November (IAEA, 2018). From Hölloch we analysed HÖL-19, which is a 415 mm-long candlestick-type stalagmite (SI Fig. 1) that grew adjacent to HÖL-7, a stalagmite that was deposited during MIS 3 in the northern part of the Hölloch system (Moseley et al., 2014), c. 800 m from the northwestern entrance and 600 m from the southern entrance (Wolf, 2006). HÖL-19 has a variable internal structure that alternates between dark brown calcite, opaque white calcite, and

cemented loam layers. For this study we concentrated on the opaque white calcite only, which is better suited for U-Th dating because of its lower detrital Th content.

Grete-Ruth-Shaft  (47.5429 N, 12.0272 E) is a small cave, 142 m in length and 39 m deep, that opens to the surface at the top of a shaft 1,435 m a.s.l. (Rittig, 2012). The mean annual temperature is c.4.5°C and the mean annual precipitation 1,327 mm (recorded at the Kufstein meteorological station located c.12 km ENE from the cave at 492 m a.s.l. (ZAMG, 2018).

Garmisch-Partenkirchen is the nearest GNIP station. A honey-brown-coloured stalagmite (HUN-14) 215 mm in length and c. 60 mm in diameter (SI Fig. 1) was recovered from the most northerly part of the system in a sheltered alcove at the base of the entrance shaft. Modern calcite precipitation seems to be absent from the cave, and appears to have been limited in the past given the low abundance of speleothems observed in the cave.

Gassel cave (47.8228 N, 13.8428 E) is a well-decorated c. 5 km-long system on the eastern side of Lake Traunsee with an

entrance at 1,225 m a.s.l. on the southern flank of the Gasselkogel mountain (1,411 m a.s.l.). The cave is orientated NNE-SSW and has a vertical extension of 156 m (Mattes, 2012, 2018). Cave air temperature is a constant 5.2 ± 0.1°C. The mean annual precipitation is 2,015 mm (recorded at the Feuerkogel meteorological station located c. 10 km west from the cave at 1,618 m a.s.l. (ZAMG, 2018)). The $\delta^{18}$O of individual precipitation events can be as low as -21.5 ‰ in December and as high as -3 ‰ in July (IAEA, 2018). The six samples (SI Fig. 1) from Gassel cave include GAS-12 (530 mm long), GAS-13

(180 mm long), GAS-22 (110 mm long), GAS-25 (215 mm long), GAS-27 (210 mm long) and GAS-29 (740 mm long). All are comprised of translucent-white/greyish calcite and were inactive at the time of collection from a chamber approximately 250 m from the entrance. GAS-29 was already broken in three parts, and here we only present the data for the 135 mm-long middle section.

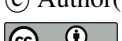



## 2.2 Analytical Methodology

The eleven stalagmites were cut in half along their growth axis and polished by a professional stone mason. Pilot dating studies guided the sample size that was needed for high precision ages. Sub-samples of between 20 to 150 mg were hand-drilled using a handheld-drill fitted with carbide burr-tipped drill bits of diameter 0.5 to 0.8 mm in a laminar-flow hood. The

cleanest, densest growth layers were targeted for sampling.

Chemical procedures and aliquot measurements were undertaken in the Trace Metal Isotope Geochemistry Laboratory at the University of Minnesota. Separation and purification of U and Th aliquots from the sub-samples was undertaken using standard methods (Edwards et al., 1987) in a clean air environment. Samples were spiked with a dilute mixed $^{229}$Th-$^{233}$U-$^{236}$U tracer to allow for correction of instrumental fractionation and calculation of U and Th concentrations and ratios. Procedural chemistry blanks were on the order of 5 – 83 ag for $^{230}$Th, 2 – 523 fg for $^{232}$Th, 73 to 171 ag for $^{234}$U, 0.2 to 1.6 pg for $^{238}$U.

Aliquots of U and Th were analysed on a ThermoFisher Neptune multi-collector inductively coupled plasma mass spectrometer (MC-ICPMS) in peak-jumping mode on the secondary electron multiplier (Shen et al., 2012).

Stable isotopes ($\delta^{18}$O and $\delta^{13}$C) were typically micro-milled at a spatial resolution of 250 μm (GAS22=200 μm ; BA7=500 μm)(SI Table 1) from the central axis of each sample (SI Fig. 2). In total 5,000 new measurements were made for this study

at the University of Innsbruck on a ThermoFisher Delta$^{plus}$XL isotope ratio mass spectrometer linked to a Gasbench II. Analytical precisions are 0.08‰ and 0.06‰ for $\delta^{18}$O and $\delta^{13}$C, respectively (1σ) (Spötl, 2011). All isotope results are reported relative to the Vienna PeeDee Belemnite standard. In addition to the main isotope track along the central axis, Hendy Tests (Hendy, 1971) were also prepared for each sample. Under this test, $\delta^{18}$O values should remain constant along a single growth layer, and there should be no relationship between $\delta^{13}$C and $\delta^{18}$O. Such tests were once used to assess whether

or not a speleothem was deposited under isotopic equilibrium (Hendy, 1971), though the preferred approach in recent years has been to reproduce the data in a second sample (Dorale and Liu, 2009). Bayesian age models were constructed for all eleven samples using OxCal version 4.2 for Poisson-process depositional models ('p sequence') and a variable 'k parameter' of 0.001 to 10 mm a$^{-1}$ (Bronk Ramsey, 2008; Bronk Ramsey and Lee, 2013). Stable isotopes and U-Th sampling positions relative to the growth axis are shown in SI Fig. 2, whilst final age models are presented in SI Fig. 3.

## 25 3 Results

### 3.1 Baschg Samples

#### 3.1.1 BA-5

The $^{238}$U concentration of the calcite in BA-5 ranges between c. 300 to 1,100 μg g$^{-1}$, and the $^{230}$Th/$^{232}$Th$_{atomic}$ ranges between c. 2000 x10$^{-6}$ to 4500 x 10$^{-6}$ (SI Table 2). The calcite is therefore moderately contaminated with detrital Th, thus resulting in

corrections to younger ages of 57-135 years, which are within the levels of uncertainty (Table 1). The resultant age model was constructed from seven U-Th ages and 279 stable isotope measurements (SI Table 1). It continuously covers the period 90.3 ± 0.3 to 85.0 ± 0.3 ka (SI Fig.3) with a resolution of 13-24 years (mean 19 years) per stable isotope measurement, and a growth rate of 10 – 20 mm ka$^{-1}$ (mean 14 mm ka$^{-1}$)(SI Table 1). $\delta^{18}$O has a 4.3 ‰ range, between -7.9 to -12.2 ‰. The relationship between $\delta^{18}$O and $\delta^{13}$C is extremely low, as indicated by the minimal variation in $\delta^{18}$O and $\delta^{13}$C along two

growth layers and linear regression analysis yielding an R$^2$ of 0.004 (SI Fig.4). Two Hendy tests reveal a range of 0.2 to 0.4 ‰ in $\delta^{18}$O and 0.2 to 0.3 ‰ in $\delta^{13}$C across single growth layers (SI Table 3).

#### 3.1.2 BA-7

BA-7 contains calcite with a $^{238}$U concentration of between c. 400 and 1,500 μg g$^{-1}$, and $^{230}$Th/$^{232}$Th$_{atomic}$ of 80 x10$^{-6}$ to 3,500 x10$^{-6}$ (SI Table 2). BA-7 is thus the 'dirtiest' of the samples analysed here. Of the16 U-Th ages used in the age model, nine



are shifted less than 1,000 years to younger ages. The age model covers the period 86.9 ± 0.3 to 80.9 ± 0.3 ka (SI Fig. 2) with a resolution of 11- 24 years (mean 15 years) per stable isotope measurement, and a growth rate of 21 – 45 mm ka$^{-1}$ (mean 34 mm ka$^{-1}$)(SI Table 1). $\delta^{18}O$ has a 4.2 ‰ range, between -7.4 to -11.6 ‰ VPDB. Linear regression analysis between $\delta^{18}O$ and $\delta^{13}C$ yields an $R^2$ of 0.3 (SI Fig.4) indicating a relationship between the two is negligible. Furthermore, four Hendy

Tests yielded a range of 0.3 to 0.4 ‰ in $\delta^{18}O$ and 0.4 to 0.5 ‰ in $\delta^{13}C$ across single growth layers (SI Table 3).

### 3.2 Schneckenloch Sample

### 3.2.1 SCH-6

The calcite of SCH-6 yields low $^{238}U$ concentrations between c. 100 and 300 µg g$^{-1}$, and $^{230}Th/^{232}Th_{atomic}$ ratios between 300 x10$^{-6}$ to 15,000 x10$^{-6}$, indicating a small incorporation of detrital Th into the sample, though the majority of the age model is

constructed from clean samples (SI Table 2). The final age model, which is constructed from seven U-Th ages, has a resolution of 6-22 years (mean 9 years) per stable isotope measurement, and a growth rate that varies between 11 to 44 mm ka$^{-1}$ (32 mm ka$^{-1}$) (SI Table 1). Six Hendy Tests reveal minimal variation in $\delta^{18}O$ (0.2 to 0.7 ‰) and $\delta^{13}C$ (0.3 to 1.0 ‰) along the respective growth layers (SI Table 3), whilst linear regression analysis yields an extremely low $R^2$ of 0.0007 (SI Fig. 4).

### 3.3 Hölloch Sample

### 3.3.1 HÖL-19

The internal morphology of HÖL-19 is variable and contains sections of clean calcite, dirty calcite, and calcified loam layers (SI Fig. 1). The youngest part of the stalagmite dates to the late Holocene and the late glacial (SI Table 2) and thus is outside the time frame for this study. Between c.95 mm and c.160 mm from the top, the stalagmite is rich in calcified loam layers

and thus is not suitable for dating. Below 160 mm there are a number of sections of clean and dirty calcite. Here we have concentrated on the cleanest part between 187.25 and 226.75 mm from the top, which yielded eight U-Th dates with c.500 to 850 µg g$^{-1}$ of $^{238}U$, and an age model between 73.6 ± 0.3 and 74.4 ± 0.2 ka (SI Table 2). The final age model has a resolution of 4-5 years (mean 5 years) per stable isotope measurement, and a growth rate that varies between 46 to 68 mm ka$^{-1}$ (mean 53 mm ka$^{-1}$). $\delta^{18}O$ has a 2.3‰ range between -8.3 to -10.6 ‰. A Hendy test at 211.75 mm from the top in the section of the

speleothem analysed here reveals minimal variation in $\delta^{18}O$ and $\delta^{13}C$ along the respective growth layer (0.3 and 0.4‰, respectively)(SI Table 3), whilst linear regression analysis yields a medium $R^2$ of 0.3 (SI Fig. 4) indicating a weak relationship between $\delta^{18}O$ and $\delta^{13}C$.

### 3.4 Grete-Ruth Sample

### 3.4.1 HUN-14

The whole length of HUN-14, which contains two hiatuses, was analysed in this study. The calcite contains between 400 to 900 µg g$^{-1}$ of $^{238}U$ and is extremely clean with $^{230}Th/^{232}Th_{atomic}$ ratios between 3,000 x10$^{-6}$ to 110,000 x10$^{-6}$. The final age model, covering the period 111.3 ± 0.3 to 102.9 ± 0.2 ka was constructed from 34 U-Th ages. The model has a resolution of 4 to 24 years (mean 10 years) per stable isotope measurement and a growth rate that varies between 11 to 57 mm ka$^{-1}$ (35 mm ka$^{-1}$). The $R^2$ between $\delta^{18}O$ and $\delta^{13}C$ is 0.2 thus indicating a negligible relationship between the two (SI Fig. 4). Four

Hendy tests also show minimal variation ($\delta^{18}O$ is 0.2 to 0.4 ‰ and $\delta^{13}C$ is 0.3 to 0.9 ‰)(SI Table 3).



### 3.5 Gassel Samples

The six Gassel samples all display similar U-Th chemical characteristics with c. 200 to 500 μg g$^{-1}$ of $^{238}$U, and extremely clean calcite with high $^{230}$Th/$^{232}$Th$_{atomic}$ ratios (SI Table 2). Correction for detrital Th is thus negligible for the final ages of these samples.

### 3.5.1 GAS-12 and GAS-13

The age models for GAS-12 and GAS-13 were constructed from 12 and 13 high-precision U-Th ages respectively, covering the intervals 81.9 ± 0.2 to 77.0 ± 0.1 ka and 80.3 ± 0.2 to 76.9 ± 0.1 ka. GAS-13 thus grew synchronous with GAS-12 for the majority of the record presented here. The age models have a resolution of 4-17 years (mean 7 years) and 3-7 years (mean 5 years) per stable isotope measurement, respectively, and growth rates that vary between 26 to 61 mm ka$^{-1}$ (mean 40 mm ka$^{-1}$) and 34 to 81 mm ka$^{-1}$ (54 mm ka$^{-1}$), respectively (SI Table 1). δ$^{18}$O has a 2.3‰ range in GAS-12, between -8.0 to -10.3 ‰, whilst GAS-13 has a slightly lower range of 1.7‰ between -8.5 to -10.2 ‰. Hendy tests that were previously carried out on both samples reveal minimal variation in δ$^{18}$O and δ$^{13}$C along the respective growth layers (Offenbecher, 2004). Linear regression analysis yields very low $R^2$ values of 0.13 for GAS-12 and 0.06 for GAS-13 (SI Fig.4) indicating minimal relationship between δ$^{18}$O and δ$^{13}$C.

### 3.5.2 GAS-22

The age model for GAS-22, covering the interval 108.0 ± 0.2 to 105.3 ± 0.1 ka, was constructed from 16 U-Th ages. The resolution per stable isotope measurement is 2 to 16 years (mean 5 years), and the growth rate varies between 13 to 100 mm ka$^{-1}$ (mean 45 mm ka$^{-1}$)(SI Table 1). δ$^{18}$O ranges from -8.3 to -11.0 ‰. Linear regression analysis between δ$^{18}$O and δ$^{13}$C yields an $R^2$ of 0.4 (SI Fig. 4) indicating a minor relationship between the two. Five Hendy tests yielded a range of 0.3 to 0.5 ‰ in δ$^{18}$O and 1.6 to 3.2 ‰ in δ$^{13}$C across single growth layers (SI Table 3).

### 3.5.3 GAS-25

The age model for GAS-25 is constructed from 17 U-Th dates and contains a hiatus. The two growth periods are modelled between 91.4 ± 0.2 to 88.2 ± 0.09 and 84.7 ± 0.1 to 83.9 ± 0.2 ka. The resolution is 4 to 8 years (mean 6 years) per stable isotope measurement, and the growth rate varies between 30 to 61 mm ka$^{-1}$ (mean 40 mm ka$^{-1}$). δ$^{18}$O has a 2.7‰ range between -7.5 to -10.2 ‰. The relationship between δ$^{18}$O and δ$^{13}$C is low, yielding an $R^2$ of 0.24 (SI Fig. 4). Six Hendy tests spread across the full length of the stalagmite yielded a range of 0.2 to 0.6 ‰ in δ$^{18}$O and 0.6 to 2.0 ‰ in δ$^{13}$C across single growth layers (SI Table 3).

### 3.5.4 GAS-27

Nine U-Th ages were used to construct the age model for GAS-27, which continually covers the period 104.9 ± 0.2 to 103.1 ± 0.2 ka with a resolution of 6 to 9 years (mean 7 years) per stable isotope measurement, and a growth rate that varies between 29 to 39 mm ka$^{-1}$ (mean 34 mm ka$^{-1}$). δ$^{18}$O has a 2.9‰ range between -8.1 to -11.0 ‰. A moderate relationship between δ$^{18}$O and δ$^{13}$C exists with an $R^2$ of 0.6 (SI Fig. 4). Five Hendy tests display a range of 0.3 to 0.8 ‰ in δ$^{18}$O and 0.3 to 4.6 ‰ in δ$^{13}$C across single growth layers (SI Table 3).

### 3.5.5 GAS-29

GAS-29 covers the interval 106.6 ± 0.2 to 104.6 ± 0.1 ka. The age model is constructed from six U-Th ages, it has a resolution of 7 to 9 years (mean 8 years) per stable isotope measurement, and displays a growth rate of 28 to 36 mm ka$^{-1}$ (mean 32 mm ka$^{-1}$). δ$^{18}$O has a 2.3 ‰ range between -8.7 to -11.0 ‰. The relationship between δ$^{18}$O and δ$^{13}$C is negligible





across the length of the record ($R^2$=0.2) (SI Fig. 4). Five Hendy tests range between 0.2 to 0.7 ‰ in $\delta^{18}$O and 0.7 to 3.0 ‰ in $\delta^{13}$C (SI Table 3).

**4 Discussion**

**4.1 Coherence and updates to NALPS19 versus NALPS**

The new records produced in this study (Fig. 2b) comprise 5,000 $\delta^{18}$O measurements dated by 145 precise U-Th ages (SI. Fig. 3, SI Table 1), which add to the NALPS chronology of Boch et al. (2011)(Fig. 2a) that comprised 7,141 $\delta^{18}$O measurements and 154 U-Th ages. Combined, the two chronologies cover the period 118.3 to 63.7 ka. Within this interval, the record is now 90 % complete, compared to 76 % in Boch et al. (2011). Where speleothems grew synchronously, major transitional events between stadials and interstadials (and vice versa) are all in agreement within uncertainty (SI Fig. 5). In

the interest of completeness and transparency, we present all $\delta^{18}$O records here, however, some samples are cleaner than others (i.e. low in detrital Th as indicated by a higher $^{230}$Th/$^{232}$Th ratio, SI Table 2) and thus deemed to be more reliably dated. The NALPS19 chronology, upon which the remainder of this discussion will be focussed, thus encompasses only the most reliably dated records from this study and Boch et al. (2011) (Fig. 2c). Furthermore, despite being 'clean', we have additionally removed large parts of the St. Beatus records (EXC3 and EXC4) from NALPS19 because of their distinctly

different $\delta^{18}$O signature to both the Gassel samples that grew at the same time and NGRIP (Boch et al., 2011). We acknowledge that there is still value in the St. Beatus records, but they are not ideal for investigations into leads, lags, and synchronicity when more comparable records exist. For further investigation into the St. Beatus samples, we encourage the reader to refer to (Boch et al., 2011). Important updates in the NALPS19 chronology (Fig. 3 and SI Fig. 6) include: (1) the addition of the cooling into GS-20: (2) a revision of the GI-20c/GS-21.1/ GI-21.1a period using multiple cleaner samples; (3)

revision of the warming into GI-21.1e and cooling into GS-22, also using a cleaner sample; and (4), revision of the interval GI-23.1 to GI-25c, which includes the addition of the previously absent GI-25a and b and a more distinctive 'shape' to GS-24 in-line with NGRIP.

**4.2 Chronological Implications**

Fig. 3 (split into 20,000 year time slices in SI Fig. 6) show the NALPS19 $\delta^{18}$O record in comparison to other iconic, well-

dated $\delta^{18}$O records from distant Northern Hemisphere regions over the interval 60 to 120 ka.  Comparison of NGRIP and NALPS19 shows that the broad-scale pattern of shifts in $\delta^{18}$O was remarkably similar during this period, including down to centennial-scale events. Differences do, however, arise when considering the timing of the events, especially when comparing to the chronologies of GICC05$_{modelext}$ (Wolff et al., 2010) and AICC2012 (Veres et al., 2013). We investigate further the timing and duration of events by applying the ramp-fitting function of Erhardt et al. (2018), which describes the

transitions via a linear ramp between two constant levels. This approach has been previously applied to the younger, late glacial portion of the NGRIP $\delta^{18}$O record (Adolphi et al., 2018); such an approach enables the consistent treatment of records. Results of the ramp-fitting are shown in Table 1, Fig. 4 and SI. Figs 7 and 8. Unfortunately results are not available for some transitions because their shape is incompatible with the transition model. Where multiple NALPS speleothem records exist for a single transition (i.e. GI-21.1a into GS-21.1, GS-23.2 into GI-23.1, and GI-24.1a into GS-24.1, see SI Fig.

6 for stratigraphic labels), we find good agreement within dating uncertainty in the timing of the start, middle, and end of the respective transitions. For the NGRIP $\delta^{18}$O record on the GICC05$_{modelext}$ chronology, we find good agreement between the start of the transitions as defined by the ramp-fitting (this study) and that of Rasmussen et al. (2014), which uses the method of taking the first data point that clearly deviates from the baseline in the $Ca^{2+}$ record rather than the noisier $\delta^{18}$O record. When comparing the different archives, good agreement is shown within uncertainty for the period between 90 to 70 ka (Fig.

4, SI. Fig. 8). This continues also for the period c.115 to 100 ka for the NALPS19, GICC05$_{modelext}$, and Asian monsoon





composite chronologies, which would all be in agreement within uncertainty (Fig. 4, SI. Fig. 8) if the GICC05$_{modelext}$ chronology was able to assign errors (Andersen et al., 2006). Only the transitions in the AICC2012 chronology appear to be too young during this interval suggesting some revision of this chronology may be needed in agreement with the findings of Extier et al., (2018).

The NALPS19 chronology presented here enables a new consideration of the duration of GS-22, which has been the basis of investigation using annual layer counting (Vallelonga et al., 2012). Vallelonga et al. (2012) proposed the duration of GS-22 to be 2,894 ± 198 years and GI-21.2 - GS-21.2 to be 350 ± 19 years (together 3,244 ± 199 years, two sigma error). This demonstrated that the GICC05$_{modelext}$ chronology was considerably under-estimated at 2,620 years (Wolff et al., 2010). The NGRIP-EDML chronology is, however, still considerably longer (GS-22, 3,625 ± 325 years; GI-21.2 - GS-21.2, 496 years;

combined 4,121 ± 325 years) than the layer-counted estimate (Capron et al., 2010b; Vallelonga et al., 2012). For NALPS19, we consider the whole period GS-22 - GI-21.2 - GS-21.2 as defined by the ramp-fitting, because the transition into GI-21.2 could not be fit by the ramp model. Ramp-fitting shows the complete GS-22 - GI-21.2 - GS-21.2 period to have occurred between 88,582 ± 123 and 84,725 ± 216 years (note NALPS19 is relative to 1950 A.D., whilst the ice core ages as presented in Vallelonga et al., (2012) are relative to 2000 A.D., which is not important when considering durations). The total duration

in NALPS19 is therefore 3,857 ± 249 years, which is in close agreement to the 4,121 ± 325 years-duration in NGRIP-EDML (Capron et al., 2010b; Vallelonga et al., 2012). NALPS19 is also within uncertainty of the 3,660 ± 263 years-duration in NALPS (Boch et al., 2011), though the NALPS19 chronology is more reliable as it is constructed from cleaner samples. Furthermore, NALPS19 is in close agreement with the duration in the Asian monsoon composite record (Kelly et al., 2006; Kelly, 2010; Cheng et al., 2016), which is 3,793 ± 805 years (88,367 ± 750 years to 84,574 ± 295 years). The speleothem

$\delta^{18}$O records from both the Alps and China therefore both support a longer-duration GS-22 - GI-21.2 - GS-21.2 period in line with NGRIP-EDML (Capron et al., 2010b; Vallelonga et al., 2012).

**4.3 NALPS 15-120 ka**

Speleothem deposits from the northern rim of the Alps now provide a near-complete record of $\delta^{18}$O variability during the last glacial period (Fig. 5; Boch et al., 2011; Moseley et al., 2014; Luetscher et al., 2015), which is remarkably similar to $\delta^{18}$O

variability recorded in the NGRIP Greenland ice core during the same period. It is hypothesised that the moisture source for both regions during the last glacial period was the North Atlantic, with the primary control on the $\delta^{18}$O of precipitation in both Greenland and the Alps being temperature. Changes to the transport pathway have, however, been proposed for the Northern Alpine speleothem record of the Last Glacial Maximum (LGM), including a southward shift in the North Atlantic storm track, which resulted in a longer transport pathway and altitude-induced Raleigh fractionation (Luetscher et al., 2015).

We now consider the full glacial Alpine speleothem $\delta^{18}$O record in further detail. In addition to the NALPS records of Boch et al. (2011), Moseley et al. (2014) and NALPS19 (this study), we also consider the Luetscher et al. (2015) record from Siebenhengste cave, which is located on the northern rim of the Alps (Fig. 1), and the Spötl et al. (2006) record from Kleegruben cave, which is located in the Central Alps to the north of the main Alpine crest (Fig. 1). We appreciate that our investigation is only a first approximation, however, a more thorough investigation would require modelling which is beyond

the scope of this study. Furthermore, given the many different factors that can influence the $\delta^{18}$O of precipitation (Dansgaard, 1964; Rozanski et al., 1993; Clark and Fritz, 1997), it would be advantageous to have stable isotope information from fluid inclusions. Unfortunately, the speleothems presented here are largely devoid of fluid inclusions (Brandstätter, unpubl. data).

Today, temperature has been shown to have the most dominant control on the $\delta^{18}$O of precipitation along the northern rim of

the Austrian Alps (Kaiser et al., 2002; Hager and Foelsche, 2015), though other factors such as changes of the moisture source, rain-out along the different transport pathways (continental effect), altitude (altitude effect), the North Atlantic Oscillation, and locally also the amount of rain (amount effect) all have some additional control (Ambach et al., 1968; Dray





et al., 1998; Kaiser et al., 2002; Hager and Foelsche, 2015; Deininger et al., 2016). To consider the effects of each of these controls on the $\delta^{18}O$ of precipitation during the last glacial period, we have first removed from the speleothem records the variability in mean ocean $\delta^{18}O$ caused by fluctuations in ice volume (Fig. 5) using a rate of 0.012 ‰ m$^{-1}$ (Rohling, 2013) and the sea-level curve of Grant et al. (2012). Details for each cave (longitude, cave depth range, catchment elevation, sample

elevation, and cave air temperature) are shown on Figs. 6a and 6b.

Mean speleothem $\delta^{18}O$ values for stadials and interstadials in the ice-volume corrected record have been calculated for each sample (Fig. 6c, SI. Fig. 9, SI. Table 4). Excluding the samples associated with the LGM (7H in MIS2) for which it is hypothesized that the transport pathway was different compared to the remainder of the last glacial (Luetscher et al., 2015), the range in mean interstadial values is 5.0 ‰ (Klaus Cramer and Siebenhengste to Kleegruben), whilst the range in mean

stadial values is slightly larger (but comparable) at 5.4 ‰ (Siebenhengste to Kleegruben)( Fig. 6c). The highest and lowest $\delta^{18}O$ values for stadials and interstadials also both come from the same caves. If we now consider the range of $\delta^{18}O$ values in different caves for a specific interstadial (and therefore roughly a specific point in time), we find that for GI-23.1, which has the largest representation in different caves, Siebenhengste displays the least depleted $\delta^{18}O$ at -7.9 ‰, followed by St. Beatus at -8.6 ‰, Gassel at -9.0 ‰, and Grete-Ruth at -9.1 ‰ (range 1.1 ‰)(Fig. 6c). A similar relationship holds true for GI-24.1

in terms of the order of caves (though Siebenhengste is missing), but $\delta^{18}O$ values are overall more depleted (Fig. 6c). There are fewer stadials that are represented by samples from different caves, however, speleothems grew in three caves during GS-23.2 and GS-24.1. These show the same trend in $\delta^{18}O$ as for interstadials with Siebenhengste being the least depleted at -9.5 and -9.6 ‰, respectively, Gassel at -10.8 and -10.7 ‰, respectively (present in two samples in GS-24.1), and Grete-Ruth at -11.5 and -11.3 ‰, respectively (1.7 ‰ range for both stadials)(Fig. 6c). During MIS 3 speleothem growth is found (at

present) in only two caves (Hölloch and Kleegruben) and the range between these two caves during both interstadials and stadials was always comparable, though the absolute values may have varied (ranges: GI-14, 3.7 ‰; GS-15.2, 3.4 ‰; GI-15.2, 3.8 ‰; GS-16.1, 3.8 ‰)(Fig. 6c).

Given the remarkable similarity with the $\delta^{18}O$ record for Greenland, which is controlled predominantly by temperature variability (Johnsen et al., 2001), we consider the dominant control on the $\delta^{18}O$ of precipitation in the northern and central

Alps during the last glacial period to have also been temperature, and the dominant moisture source to have been the North Atlantic (as both are today). Considering the distance from the North Atlantic (continental effect), which increases with longitude, we find that no correlation existed between mean $\delta^{18}O$ and distance from the North Atlantic (Fig. 6d: IS: $R^2$=0.10, n 33; S: $R^2$=0.04, n 26). Such an approach, however, does not allow for changes in $\delta^{18}O$ with time, we therefore consider the specific time periods of GI-23.1, GS-23.2, GI-24.1, and GS-24.1, for which the largest number of caves are represented (Fig.

6e). In all instances, $\delta^{18}O$ becomes increasingly depleted with increasing distance from the North Atlantic; a medium correlation is displayed for GI-23.1 ($r^2$=0.64, n=4), GS-23.2 ($r^2$=0.63, n=3), GS-24.1 ($r^2$=0.57, n=4, two samples for Gassel), and a lower correlation during GI-24.1 ($r^2$=0.16, n=3). This trend of decreasing $\delta^{18}O$ with increasing distance from the source would be expected with progressive rainout, in particular in a cold climate.

Today, spatial variability of the $\delta^{18}O$ of precipitation in the Austrian Alps is highly dependent on altitude (Hager and

Foelsche, 2015). For a given location, however, Ambach et al. (1968) argued that the altitude effect cannot be the result of a difference in condensation temperature, because the condensation level should be approximately the same. Instead they argued that the altitude effect is largely caused by $^{18}O$ enrichment caused by evaporation from falling rain drops. The effect is most prominent in the summer months when precipitation falls as rain, rather than the winter months when more precipitation falls as snow from which fractionation by evaporation does not occur (Ambach et al., 1968). Considering the

relationship between catchment elevation and $\delta^{18}O$, we once again consider the specific time periods of GI-23.1 ($r^2$=0.49, n=4), GI-24.1 ($r^2$=0.67, n=3), GS-23.2 ($r^2$=0.79, n=3), GS-24.1 ($r^2$=0.74, n=4 (Gassel has 2 samples)), which all display medium to strong correlations (Fig. 6f). With the exception of GI-24.1, which does not contain the high-elevation Siebenhengste site, $\delta^{18}O$ increases with elevation, i.e. opposite to what would be expected if the 'altitude effect' had a strong

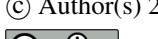


control. Removal of the high-elevation Siebenhengste site from these three time periods reverses the trend to decreasing $\delta^{18}O$ with increasing catchment elevation (Fig. 6g; GI-23.1 $r^2$=1.0, n=3; GS-23.2 $r^2$=not applicable because n=2; GS-24.1 $r^2$=1.0, n=3 (Gassel has 2 samples)). With such a spread of mean $\delta^{18}O$ values in different caves at different times, it is difficult to make firm conclusions, though we offer some hypotheses based on the available data. As discussed, we have shown that

$\delta^{18}O$ trends towards increasingly depleted $\delta^{18}O$ with increasing distance from the dominant moisture source for a given time period (Fig. 6e). Despite this, there is some variability overprinted on top of this trend. For instance, mean $\delta^{18}O$ values for Grete-Ruth are consistently more depleted than for Gassel cave despite Grete-Ruth being closer to the North Atlantic. Grete-Ruth is, however, located at a higher altitude than Gassel cave, and accordingly displays more depleted mean $\delta^{18}O$ values (Fig. 6f). St. Beatus and Siebenhengste, which are located close to one another and closest to the North Atlantic, tend to

display less depleted $\delta^{18}O$ values than Grete-Ruth or Gassel cave (Fig. 6d/6e); this would be expected based on the continental effect, however, mean $\delta^{18}O$ values for St. Beatus and Siebenhengste are reversed during GI-23.1 (Fig. 6e) in comparison to what one would expect for the altitude effect. Given that the condensation level and therefore condensation temperature are expected to be same in the same location, one must consider the reason for the difference in $\delta^{18}O$ for these two caves. Given that the lower-elevation sites follow the expected altitude-induced trend of decreasing $\delta^{18}O$ with increasing

elevation (Fig. 6f), it is possible that the $\delta^{18}O$ values in the high-elevation Siebenhengste cave are biased towards a summer signal. Such a bias could for instance be caused by wind erosion resulting in relocation of the isotopically-light winter snow, a process that has been well-documented at various Alpine sites (Ambach et al., 1968; Bohleber et al., 2013; Hürkamp et al., 2019). Alternatively, if firn developed above Siebenhengste during GI-23.1, then this would also limit the input of isotopically-light precipitation in the recorded signal. At present, however, there is no evidence to either support or reject the

hypothesis of firn above Siebenhengste during MIS 5. An alternative theory to explain the discrepancy between mean $\delta^{18}O$ values for lower-elevation sites versus the high-elevation Siebenhengste could be that Siebenhengste records the full annual signal and the lower-elevation sites are winter biased towards isotopically-lighter values. Such a mechanism could be explained by considering evapotranspiration. At higher-elevation sites that are absent of vegetation, the full annual signal of precipitation is likely to recharge the aquifer. In contrast, at lower-elevation vegetated sites, summer recharge to the aquifer

may be reduced by evapotranspiration processes resulting in a winter bias in the recorded $\delta^{18}O$ signal. Such an explanation is not easily supported by our data, since a general trend of decreasing $\delta^{18}O$ values with increasing elevation does exist for the lower-elevation sites and only Siebenhengste appears anomalous (Fig. 6f). However, the effects of evapotranspiration may only be overprinted on top of an otherwise altitude-dominant signal. We also acknowledge that in the late glacial, Siebenhengste samples do record much lighter $\delta^{18}O$ signals (Fig. 6c) though at present we do not have a full explanation for

this difference. The high mean $\delta^{18}O$ values for the high-elevation Siebenhengste cave do appear to be somewhat anomalous within this particular dataset. For instance, during MIS 3, the $\delta^{18}O$ values for the high-elevation Kleegruben cave (Fig. 6f), are more depleted than for Hölloch cave. This would be expected based on both the altitude effect and given that Kleegruben is located further from the North Atlantic than Hölloch. Given that this cave was overlain by a warm-based glacier at this time (Spötl et al., 2006), it is possible that its dripwater at that time reflected more of the annual signal. The final high-

elevation cave presented here is Klaus Cramer (Fig. 6a) though speleothems from this cave were not growing synchronous with speleothems from any other caves (Fig. 6c) thus limiting our ability to know whether the Klaus Cramer $\delta^{18}O$ signal is in agreement with what one would expect for its altitude. Despite this, when comparing the Klaus Cramer GI-19.2 and GS-19.2 $\delta^{18}O$ signal with mean $\delta^{18}O$ signals for other interstadials and stadials, it would seem that $\delta^{18}O$ values are depleted as expected for the stadial but not the interstadial (Fig. 6c).

The state of the NAO is also known to exhibit control on precipitation characteristics in the Alps, particularly during winter (Wanner et al., 1997), though in comparison to other European sites the relationship is not as strong (Baldini et al., 2008). For instance, positive NAO indices indicate strong zonal flow, result in low precipitation amounts and high temperatures during winter, whilst the opposite is true for NAO negative or low phases (Wanner et al., 1997; Beniston and Jungo, 2002;





Casty et al., 2005). Such controls are strongest at high elevation (Beniston and Jungo, 2002), and sometimes found to be unstable, meaning that at times the state of the NAO is found to be uncorrelated with temperature or precipitation amount (Casty et al., 2005). When a relationship does exist, the $\delta^{18}O$ of precipitation is found to be positively correlated with the NAO index (Kaiser et al, 2002), though this is perhaps unsurprising given the positive relationship between the NAO index

and temperature. When considering the last glacial period, the LGM NAO was characterized by four distinct centres of action that resulted in distinct differences in atmospheric circulation and internal variability compared to today, namely a shift in winds from a strong westerly-dominated flow to a southwest-northeast orientated flow (Justino and Peltier, 2005). At present, the timing of the shifts between two and four centres of action is not known, hence limiting our ability to investigate the relationship between the NAO phase and the $\delta^{18}O$ of precipitation during MIS 3 to 5 further.

**4.4 Stadial-level centennial-scale cold events**

The recognition of centennial to millennial-scale climate events, such as 'precursors' to interstadials and within-interstadial depletions in $\delta^{18}O$ (Capron et al., 2010a), led to the designation of a new stratigraphic framework for the Greenland ice cores over the last glacial period (Rasmussen et al., 2014). Typically, a 'precursor-event' is a feature of a stadial termination; this includes GS-16.2, 17.2, 21.2 and 23.2 (Fig. 7)(Capron et al., 2010a; Rasmussen et al., 2014). It is characterised in both

Northern Alpine speleothems and Greenland ice cores by a rapid increase in $\delta^{18}O$ from stadial to interstadial conditions. The event is short-lived, lasting a maximum of a few centuries before $\delta^{18}O$ returns to near-stadial conditions for another few decades to centuries, followed by the main transition into the interstadial. $[Ca^{2+}]$ in the Greenland ice cores varies almost simultaneously with these $\delta^{18}O$ changes, where increases in $[Ca^{2+}]$ are associated with depletions in $\delta^{18}O$ and vice versa. Changes to $[Ca^{2+}]$ are interpreted to reflect changes in dust concentration caused by changes in dust source conditions and

transport pathways indicative of regional-to-hemispheric-scale circulation changes (Rasmussen et al., 2014). In comparison, 'within-interstadial' climate perturbations are characterised in general by smaller-amplitude depletions in $\delta^{18}O$ that typically do not reach stadial values, and are often of shorter duration than the stadial-termination reversals. $[Ca^{2+}]$ also varies in-tune with 'within-interstadial' $\delta^{18}O$ fluctuations, but similarly does not reach full stadial values. Such characteristics appear to be consistent in $\delta^{18}O$ records from both Greenland ice cores and Northern Alpine speleothems. The exception to such typical

'within-interstadial' cold perturbations is the event at 107.5 ka in the NALPS19 chronology, which is designated GS-24.2 in the NGRIP stratigraphy (Rasmussen et al., 2014). This drop in $\delta^{18}O$ to stadial values occurred 978 years after the start of the interstadial, thus firmly making it a 'within-interstadial' event rather than one associated with a stadial termination. At present, the 107.5 ka-event (GS-24.2) is the only centennial-scale $\delta^{18}O$ event of such extreme amplitude occurring during an interstadial that is recognised in both Greenland and Northern Alpine records. Because of this, it has been likened to the 8.2

ka cold event that occurred in the early Holocene (Alley et al., 1997; Capron et al., 2010a). Still, the $\delta^{18}O$ excursion of the 8.2 ka event did not reach near-stadial values in NGRIP as GS-24.2 did, thus highlighting some differences between these two warm-interrupting cold reversals. In addition, Rasmussen et al. (2014) liken the 'within-interstadial' GS-24.2 cold perturbation to stadial termination events GS-16.2 and GS-17.2. Both the similarities between GS-24.2 and the 8.2 ka event, as well as with GS-16.2 and GS-17.2, suggest that such abrupt climate variability is not critically influenced by the size of

the Greenland ice sheet (Capron et al., 2010a; Rasmussen et al., 2014).

During the deglacial and early Holocene, large-scale meltwater events are widely suggested as being responsible for causing some short-term climate reversals through the weakening of Atlantic meridional overturning circulation (AMOC) (e.g., Broecker et al, 1994; Teller et al., 2002; Clark et al., 2001, 2004). Such cold reversals thought to be triggered by meltwater events include the Older Dryas at 14 ka (GI-1d, Rasmussen et al., 2014), the Preboreal Oscillation at 11.4 ka (e.g., Johnsen et

al., 1992; Björck et al. 1996; Fischer et al., 2002), the 9.3 ka event (Fleitmann et al., 2008; Yu et al., 2010), and the 8.2 ka event (Alley et al., 1997). In contrast though, not all freshwater injections led to cold events, and not all cold events were caused by freshwater injections (Stanford et al., 2006). For instance, both the Younger Dryas and Heinrich events occurred




during times of already-colder sea surface temperatures and weakened AMOC, indicating that the input of freshwater from the iceberg armadas was not the initial cause of the AMOC slowdown (e.g., Hall et al., 2006; Henry et al., 2016).

In the case of the centennial-scale cold reversals of GS-16.2, GS-17.2, GS-21.2, GS-23.2 and GS-24.2 (Fig. 7), a possible mechanism for each of these events could be similar to the meltwater-triggered cold reversals of the deglacial. This

hypothesis is supported when considering that events GS-17.2, GS-21.2, and GS-24.2 occurred shortly following episodes of rapid sea-level rise, which were in excess of 12 m ka$^{-1}$ in the high-resolution record of Grant et al. (2012)(Fig. 7). Such rapid sea-level rise does not appear to have occurred prior to GS-23.2, though closer inspection of the sea-level curve shows that following the rise prior to GS-24.2, sea levels had remained elevated and underwent a series of rapid oscillations that are smoothed out in the rate-of-change curve (Fig. 7). Likewise, GS-16.2 did not occur coincident with an episode of sea-level

rise, but did occur shortly after the rise associated with GS-17.2 (Fig. 7). Additionally, the rapid rises in sea level each began at times of increased ice-rafted debris (IRD) in the North Atlantic (McManus et al., 1999, on U-Th timescale), weakened AMOC and increased ice volume as indicated by high benthic $\delta^{13}C$ and $\delta^{18}O$ values, respectively, as well as pluvial periods in Brazil caused by a southward shift of the intertropical convergence zone (ITCZ) (Wang et al., 2004)(Fig. 7). In the late glacial, such episodes are associated with Heinrich events (Wang et al., 2004). Furthermore during glacial terminations, the

sequence of events has been shown to include a Heinrich event, followed by short-lived warming, then a millennial-scale return to cold conditions, and finally the transition to the interglacial (Cheng et al., 2009). Though on shorter timescales, the pattern of events during these specific stadial terminations is similar to the pattern of events during glacial terminations. The 'stadial-termination' oscillations of GS-16.2, GS-17.2, GS-21.2 and GS-23.2 can therefore be considered as being akin to the meltwater-triggered Preboreal Oscillation, which occurred shortly following the warming at the end of the Younger Dryas

stadial during a time when considerable volumes of ice still existed similar to the early glacial. These stadial termination reversals during the early glacial period are therefore not so much warming events that punctuate cold periods (Capron et al., 2010a), but rathermore small-scale terminations that failed due to freshwater influx. On the other hand, GS-24.2, which occurred nearly 1,000 years after warming occurred, is more similar to the Older Dryas in which a cold event punctuated a warm interval.

## 25  5 Conclusions

In this paper, we present the most recent chronology, named NALPS19, for $\delta^{18}O$ variability as recorded in speleothems that grew during the last glacial period between c. 15 and 120 ka along the northern rim of the Alps. In particular, we have updated the record between 63.7 to 118.3 ka, using eleven cleaner, more accurately and precisely dated samples from five caves. Over the 63.7 to 118.3 ka interval, the record is now 90% complete. Ramp-fitting analysis of the main transitions

between stadials and interstadials and vice-versa shows synchronicity within dating uncertainty with equivalent shifts in the NGRIP $\delta^{18}O$ record on the GICC05$_{modelext}$ chronology and the Asian Monsoon composite speleothem record. Major differences do, however, arise in comparison to NGRIP on AICC2012. Furthermore, we suggest that the highly-debated GS-22 - GI-21.2 - GS-21.2 interval had a duration of 3,857 ± 249 years, which is in closer agreement with the 4,121 ± 325 years of NGRIP-EDML (Capron et al., 2010b) and the 3,793 ± 805 years of the Asian monsoon composite (Kelly et al., 2006;

Kelly, 2010; Cheng et al., 2016). Preliminary investigation of the trends in mean $\delta^{18}O$ as recorded in the NALPS speleothems for different interstadial and stadials reveals that for a given time period, as expected, $\delta^{18}O$ becomes more depleted with increasing distance from the source and increasing elevation. Exceptions are found at some high-elevation sites, which appear to record a summer bias. Finally, our accurate and precise chronology enables a deeper investigation of centennial-scale cold reversals that occurred either as 'precursor events' (i.e., GS-16.2, GS-17.2, GS-21.2, GS-23.2; Capron et

al., 2010a) or during interstadials (i.e. GS-24.2). Each of these events occurred shortly following rapid rises in sea level of over 12 m kyr$^{-1}$ (Grant et al., 2012) that occurred coincident with IRD events (McManus et al., 1999) and shifts in the ITCZ causing speleothem growth in Brazil (Wang et al., 2004). We therefore propose that these centennial-scale cold reversals are





products of freshwater discharge into the North Atlantic during times of moderate ice sheet size, which caused a slowdown of the AMOC and associated atmospheric cooling, similar to deglacial events such as the Preboreal Oscillation or Older Dryas.

**Data availability**

The stable isotope data both on distance along growth axis and OxCal age models are available at the US National Oceanic and Atmospheric Administration (NOAA) data center for paleoclimate (speleothem site) at the following address: TBC

**Author contribution**

GM undertook the majority of the U-Th analyses, interpreted the data, and wrote the manuscript. CS conceived the project. SB undertook additional U-Th analyses, prepared and ran Hendy tests and stable-isotope samples. TE developed and ran

ramp-fitting models. ML provided data. RLE provided analytical U-Th facilities. All authors directly contributed to the manuscript through discussion or writing.

**Competing interests**

The authors declare that they have no conflict of interest.

**Acknowledgements**

This work was funded primarily by FWF grant P222780 to CS, with a smaller contribution from FWF grant T710-NBL to GM. TE acknowledges the long-term financial support of ice-core research by the Swiss National Science Foundation (SNSF) and the Oeschger Center for Climate Change Research. We thank J. Nissen, A. Berry and A. Min for analysis of U-Th aliquots; Y. Lu, P. Zhang, and X. Li for laboratory management; M. Wimmer for her assistance in the stable isotope lab, and; J. Degenfelder for production of Fig. 1. We also thank PHC Amadeus 2018 Project 37910VD for supporting workshops

where useful discussions were held that contributed to the interpretation of this manuscript.

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

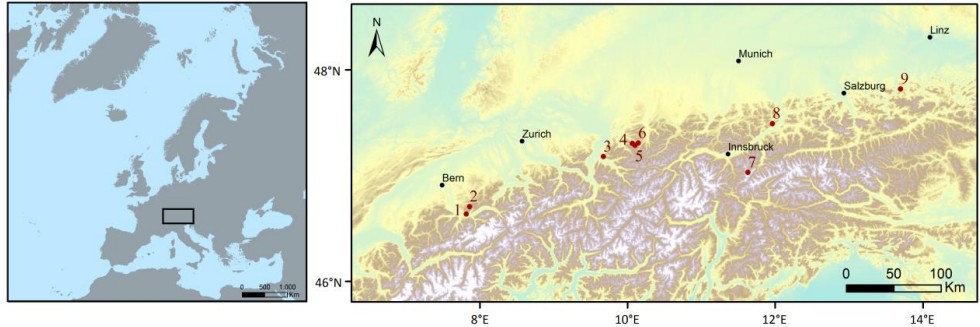

**Figure 1: Map of cave sites discussed in text. 1. St. Beatus cave; 2. Siebenhengste cave; 3. Große Baschg cave; 4. Schneckenloch**
**cave; 5. Klaus Cramer cave; 6. Hölloch cave; 7. Kleegruben cave (part of wider discussion on isotopic controls); 8. Grete-Ruth**
**Shaft; 9. Gassel cave.**





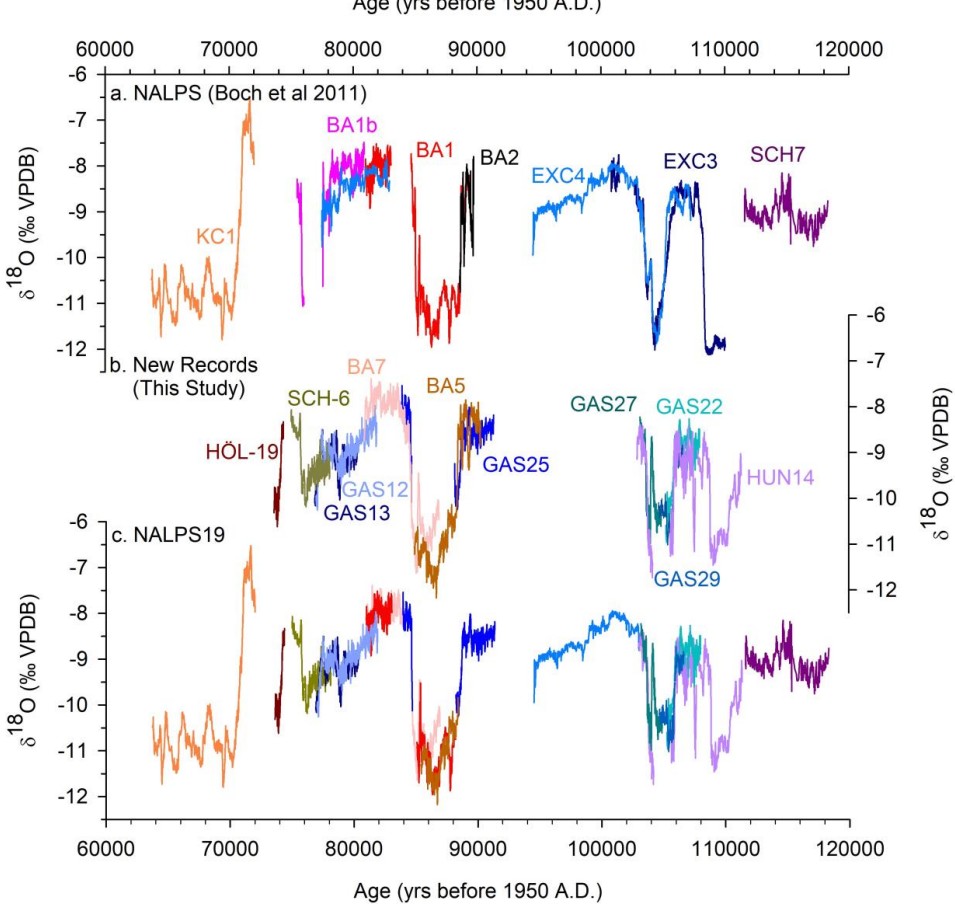

**Figure 2: NALPS δ¹⁸O speleothem records a. Original NALPS record of Boch et al. (2011); b. new records from this study; c. the most reliable records of Boch et al. (2011) and this study combined to form NALPS19.**





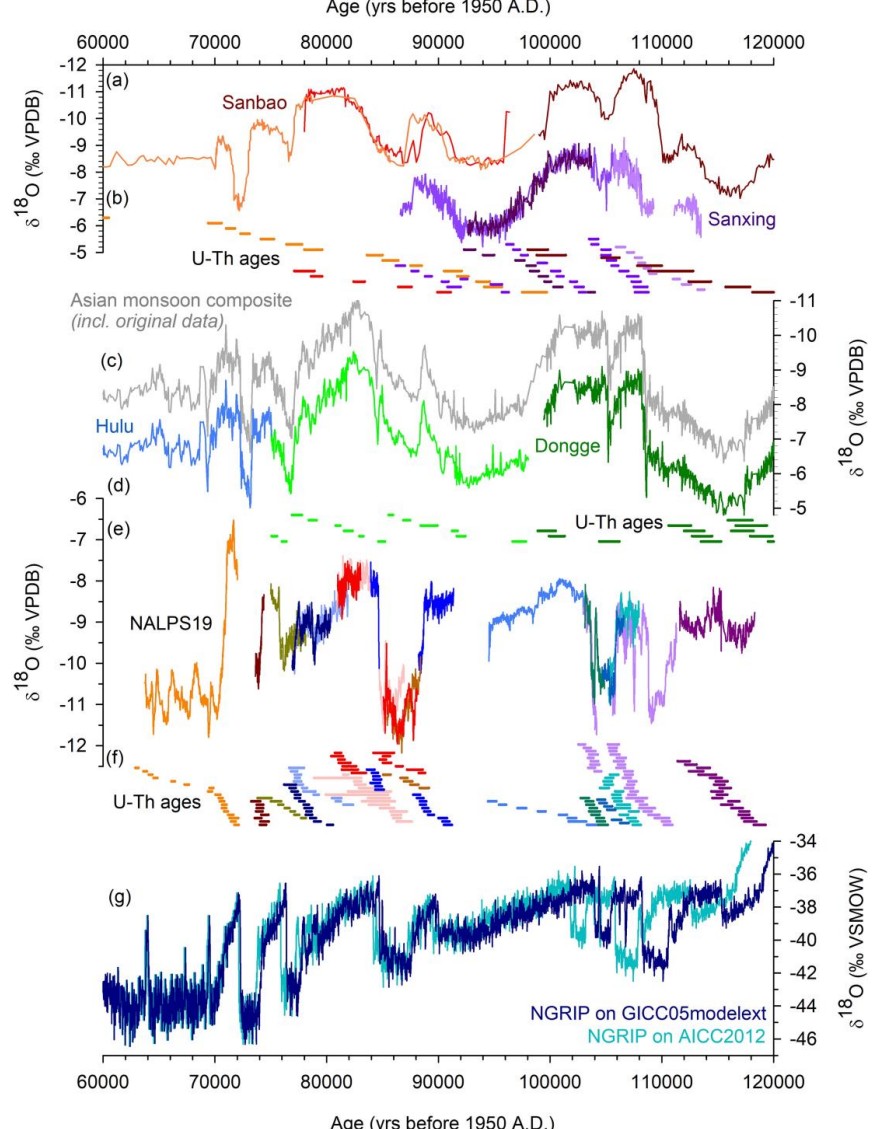

**Figure 3: NALPS19 δ¹⁸O record versus other well-dated δ¹⁸O records. (a) Chinese speleothem δ¹⁸O records from Sanbao (Wang et al., 2004) and Sanxing caves (Jiang et al., 2016). (b) 2σ range of U-Th ages used to produce (a) are colour-coded the same as (a). (c) Asian monsoon composite record (Cheng et al., 2016) as well as the original data from which it was constructed (Dongge; Kelly et al., 2006; Kelly, 2010). (d) 2σ range of U-Th ages used to produce (c) are colour-coded the same as (c). (e) NALPS19 record (this study). (f) 2σ range of U-Th ages used to produce (e) are colour-coded the same as (e). for (e) colour-coded the same. (g) NGRIP records on the GICC05modelext chronology (Svensson et al., 2008; Wolff et al, 2010) and AICC2012 chronology (Veres et al., 2013). To see this graph split into 20,000 year slices and with the NGRIP nomenclature (Rasmussen et al., 2014), see SI Fig. 6.**



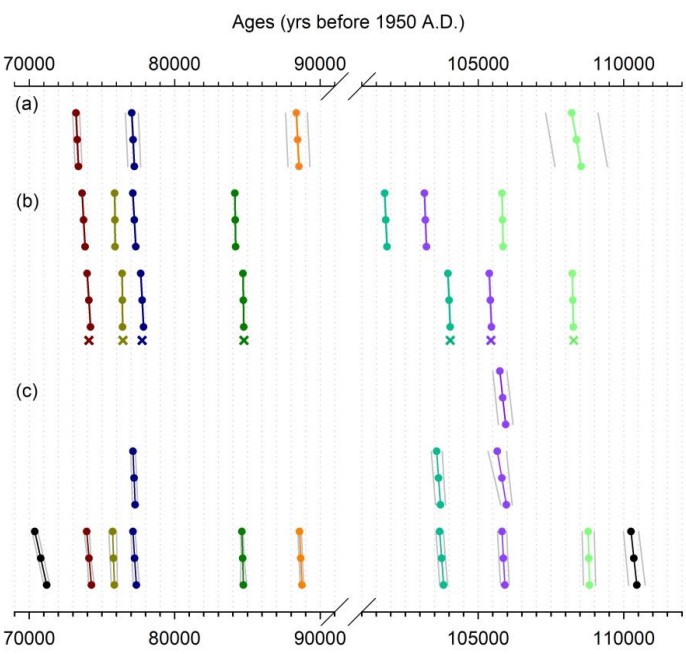

**Figure 4: The timing of transitions as defined by the ramp-fitting model of Erhardt et al. (2018) in (a) Asian monsoon composite speleothem δ18O record (Kelly et al., 2006; Kelly, 2010; Cheng et al., 2016), (b) NGRIP δ18O record on AICC2012 chronology (top) (Veres et al., 2013) and GICC05_modelext chronology (bottom) (Wolff et al, 2010), and (c) NALPS19 (this study). Colours are used to**
5    **denote the same transition in each record respectively. Circles mark the start, middle and end of each transition. Grey vertical bars mark the typical 2σ U-Th dating uncertainty. The GICC05_modelext chronology does not contain uncertainties in this time period (Wolff et al., 2010) whereas the AICC2012 chronology has extremely large uncertainties of up to 6,000 years (Veres et al., 2013). Crosses on (b) mark the start of transitions as defined in Rasmussen et al. (2014).**

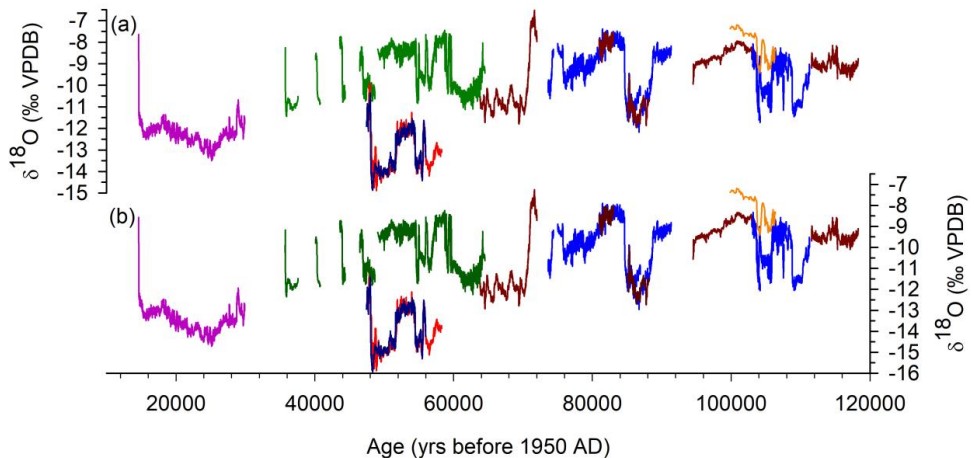

10   **Figure 5: Speleothem δ18O records from the northern rim and central European Alps. (a) Original records: pink (Luetscher et al., 2015), green (Moseley et al., 2014), red and dark blue (Spötl et al., 2006), dark red (Boch et al., 2011 contained in NALPS19), medium blue (new record in this study), orange (Luetscher see SI Table 4 and SI Fig. 10). (b) δ18O records corrected for δ18O variability as a result of changing ice volume. Colour codes the same as in (a).**




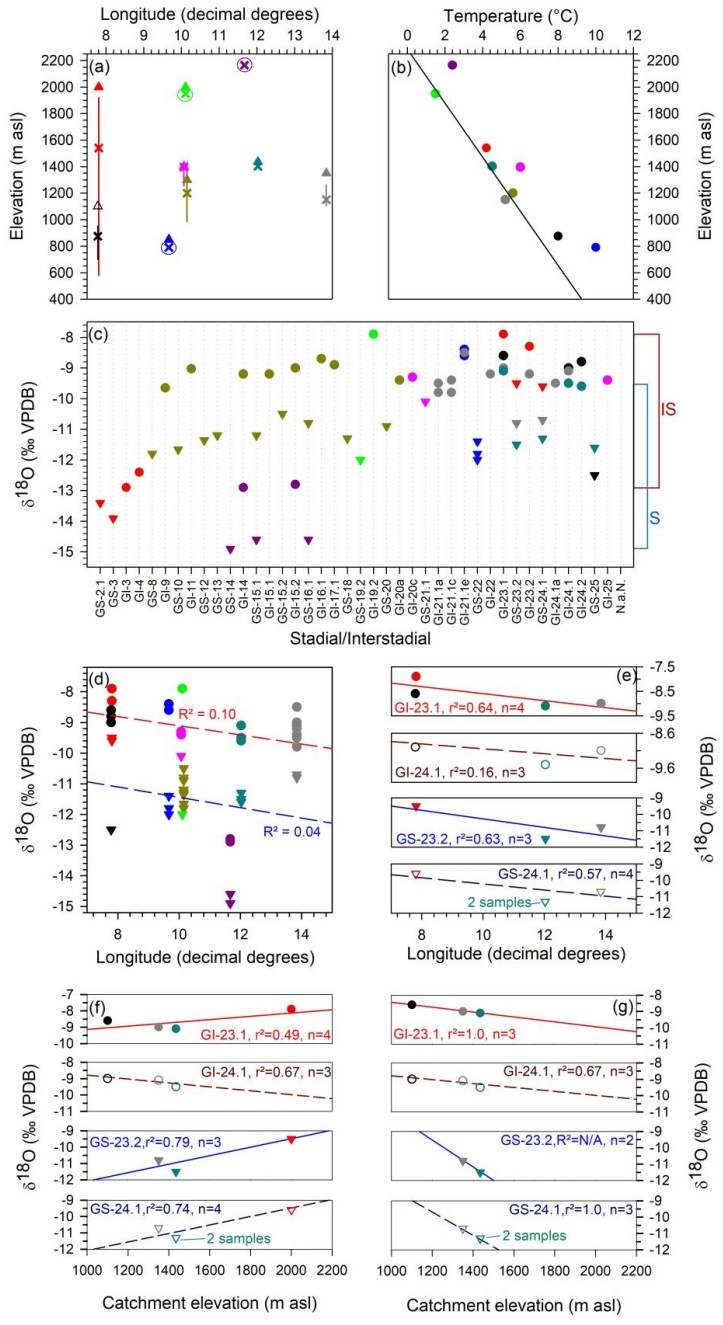

**Figure 6: Mean δ¹⁸O values for specific interstadials and stadials (see SI Table 4 and SI Fig. 9). (a) Cave information. Catchment elevation (triangle). Sample location (cross). Cave depth range (vertical bar). Where cave depth is too small to see on this scale, the sample location is circled. Siebenhengste (red), St. Beatus (black), Baschg (blue), Klaus Cramer (green), Schneckenloch (pink),**
5 **Hölloch (dark yellow), Kleegruben (dark pink), Grete-Ruth (cyan), Gassel (grey). (b) Cave temperature relative to elevation. Colours are the same as in (a). Black solid line represents the average lapse rate for the Eastern Alps based on instrumental data 1981-2010 (source: ZAMG). (c) Mean δ¹⁸O values for specific stadials (triangles) and interstadials (circles). Colours are the same as in (a). IS = interstadial range. S = stadial range. (d) All mean δ¹⁸O values plotted relative to longitude. Colours are the same as in (a). Interstadials (circles). Stadials (triangles). (e) Mean δ¹⁸O values for specific time periods plotted relative to longitude.**
10 **Colours are the same as in (a). Interstadials (circles). Stadials (triangles). (f) All mean δ¹⁸O values plotted relative to catchment elevation. Colours are the same as in (a). Interstadials (circles). Stadials (triangles). (g) Mean δ¹⁸O values for specific time periods plotted relative to catchment elevation. Colours are the same as in (a). Interstadials (circles). Stadials (triangles).**



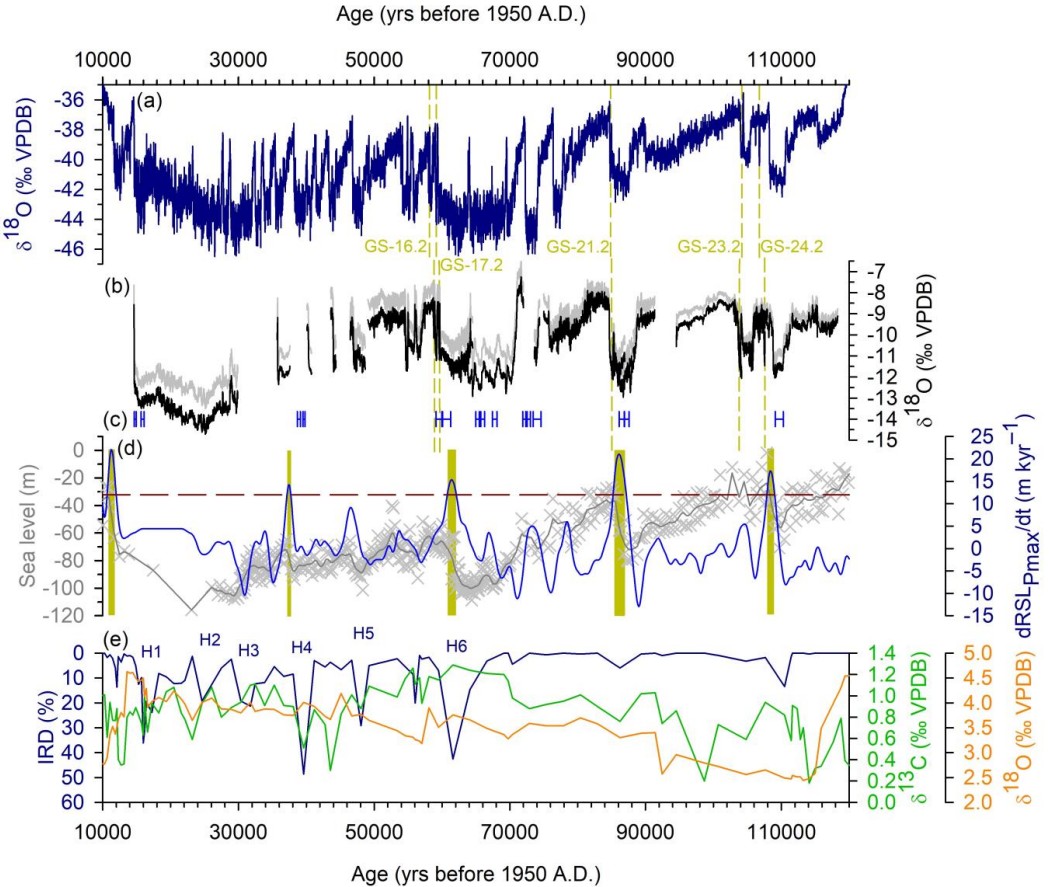

**Figure 7: (a) NGRIP δ¹⁸O on GICC05_modelext (Wolff et al., 2010). (b) NALPS19 δ¹⁸O uncorrected for variability in ocean δ¹⁸O (grey), corrected for variability in ocean δ¹⁸O (black). (c) Growth periods in Brazilian speleothem (Dark blue) (Wang et al., 2004). Sea-level variability (Grant et al., 2012). Relative sea-level data (grey crosses). Maximum-probability relative sea-level (grey line). Rate of sea-level change (blue line). Rate of 12 m kyr⁻¹ indicated by horizontal red line. Peaks of sea-level change in excess of 12 m kyr⁻¹ indicated by yellow bars. (e) Ice-rafted debris (dark blue), benthic δ¹³C (green), and planktic δ¹⁸O (orange) from ODP980 on Hulu U-Th age scale (McManus et al., 1999; Barker et al., 2011).**




**Table 1. Results of the ramp-fitting model runs for NALPS19 (this study), NGRIP on GICC05modelext (Wolff et al., 2010), AICC2012 (Veres et al., 2013), and the Asian monsoon composite (Kelly et al., 2006; Kelly, 2010; Cheng et al., 2016). All ages are reported relative to 1950 A.D. Uncertainties given are modelling uncertainties as marginal posterior standard deviations. Uncertainties in parentheses are associated U-Th or OxCal uncertainties.**

| Transition | Sample | NALPS19 Start | NALPS19 Mid-point | NALPS19 End | GICC05$_{modelext}$ Start | GICC05$_{modelext}$ Mid-point | GICC05$_{modelext}$ End | AICC2012 Start | AICC2012 Mid-point | AICC2012 End | Asian Composite Start | Asian Composite Mid-point | Asian Composite End |
|---|---|---|---|---|---|---|---|---|---|---|---|---|---|
| GI-19.2-GS-19.2 | KC1 | 71104 ±28 (210) | 70859 ±19 (200) | 70615 ±34 (190) | | | | | | | | | |
| GI-20a-GS-20 | HÖL-19 | 74262 ±18 (189) | 74146 ±12 (130) | 74031 ±24 (138) | 74219 ±23 | 74101 ±14 | 73984 ±22 | 73846 ±21 | 73741 ±13 | 73635 ±20 | 73389 ±24 (240) | 73306 ±13 (240) | 73222 ±20 (240) |
| GS-21.1-GI-20c | SCH-6 | 75852 ±23 (213) | 75795 ±11 (240) | 75738 ±10 (266) | 76417 ±14 | 76403 ±5 | 76390 ±10 | 75904 ±15 | 75890 ±6 | 75876 ±11 | | | |
| GI-21.1a-GS-21.1 | GAS-12 | 77372 ±30 (146) | 77251 ±18 (177) | 77129 ±24 (217) | 77865 ±27 | 77763 ±15 | 77661 ±27 | 77332 ±27 | 77229 ±16 | 77127 ±26 | 77224 +45 (440) | 77138 ±27 (440) | 77053 ±41 (440) |
| GI-21.1a-GS-21.1 | GAS-13 | 77296 ±41 (158) | 77207 ±22 (176) | 77119 ±27 (173) | | | | | | | | | |
| GS-21.2-GI-21.1e | BAS-7 / GAS-25 | 84725 ±16 (216) | 84671 ±8 (100) | 84618 ±15 (90) | 84751 ±10 | 84724 ±4 | 84697 ±6 | 84194 ±11 | 84166 ±4 | 84137 ±7 | | | |
| GI-22a-GS-22 | GAS-25 | 88747 ±17 (117) | 88664 ±14 (118) | 88582 ±32 (123) | | | | | | | 88540 ±60 (750) | 88454 ±25 (750) | 88367 ±39 (750) |
| GS-23.2-GI-23.1 | HUN-14 | 103814 ±20 (136) | 103745 ±9 (129) | 103676 ±22 (131) | 104042 ±15 | 104001 ±6 | 103961 ±11 | 101868 ±14 | 101829 ±5 | 101791 ±11 | 103734 ±153 (800) | 103694 ±149 (800) | 103653 ±150 (800) |
| GS-23.2-GI-23.1 | GAS-27 | 103705 ±22 (172) | 103640 ±9 (178) | 103575 ±19 (188) | | | | | | | | | |
| GI-24.1a-GS-24.1 | HUN-14 | 105916 ±18 (149) | 105868 ±10 (154) | 105820 ±18 (159) | 105455 ±37 | 105418 ±16 | 105382 ±17 | 103226 ±37 | 103189 ±16 | 103153 ±18 | | | |
| GI-24.1a-GS-24.1 | GAS-22 | 105971 ±28 (199) | 105814 ±23 (253) | 105657 ±47 (318) | | | | | | | | | |
| GI-24.1a-GS-24.1 | GAS-29 | 105944 ±17 (252) | 105845 ±8 (248) | 105745 ±13 (240) | | | | | | | | | |
| GS-25-GI-24.2 | HUN-14 | 108825 ±6 (210) | 108801 ±4 (210) | 108778 ±8 (210) | 108271 ±12 | 108251 ±5 | 108231 ±6 | 105855 ±11 | 105839 ±4 | 105819 ±6 | 108538 ±44 (900) | 108377 ±31 | 108217 ±51 |
| GI-25a-GS-25 | HUN-14 | 110450 ±44 (284) | 110350 ±23 (274) | 110251 ±34 (263) | | | | | | | | | |