# Peer review of "NALPS19: Sub-orbital scale climate variability recorded in Northern Alpine speleothems during the last glacial period"

_Climate of the Past, 2019_

## Referee Comment (RC1) · Anonymous Referee #1 · 27 May 2019

1- SUMMARY AND GENERAL COMMENTS:

The study by G. Moseley and collaborators presents 11 new radiometrically-dated speleothem records from five caves located on the northern rim of the Alps. Those records are used to propose a revision of the composite NALPS calcite $\delta18O$ curve published in Boch et al. (2011) covering the time interval ∼120-60 ka. Using those new speleothem records and comparing them to Asian speleothem $\delta18O$ records and Greenland NGRIP ice core $\delta18O$ records, the authors investigate the coherence and possible lead and lags in the timing of the abrupt stadial-interstadial transitions. To do so, they use a recent statistical tool to objectively define the onset and the end of

the abrupt changes in the different records. They evidence a good agreement within the dating uncertainties between the NALPS calcite $\delta18O$, the radiometrically-dated Asian speleothem composite $\delta18O$ and the NGRIP ice $\delta18O$ record when the later is displayed on the GICC05modelext chronology (Wolff et al. 2010). The authors discuss the relative influence of the different control factors on the NALPS calcite $\delta18O$ record. Finally, the new NALPS composite $\delta18O$ curve provides further evidence that the sub-millennial-scale climate variability identified in the NGRIP ice $\delta18O$ record is also visible in continental records from the Northern Alps. Taking advantage of their precisely-dated record, the authors propose a possible mechanism for triggering this sub-millennial-scale variability, involving ice sheet melting impacting the intensity of the Atlantic Meridional Overturning Circulation (AMOC).

This paper presents substantial new material and it will be of great interest for the speleothem community and to the extended paleoclimate community. It is thus well within the scope of Climate of the Past. However, many aspects of the paper related both to the form and the content need improvements and clarifications. Hence, I recommend that major revisions are being done before it is considered for publication.

Before I describe my comments in the following, I would like to mention that my limited expertise on the technicalities for dating speleothem records prevents me from providing a careful evaluation of the methodology described here. I thus hope that this important aspect of the submitted work will be carefully addressed by expert(s) in the field.

My first general comment is that I find that the manuscript is unbalanced as some sections of the paper should be shortened while others would deserve to be expended. In particular:

- Section 2.1 presenting the cave sites and speleothems could be shortened. Instead of a long and tedious text to read, it would be helpful to have an associated table in the main manuscript that summarizes at least some of the information provided in the

text e.g. cave name, coordinates, elevation, air temperature and precipitation rate, associated sample acronyms, length of the samples. . .).

- The discussion is also difficult to follow in Section 4.3, and the take-home messages hard to identify. The authors are investigating and discussing the roles of the different potential control factors on the calcite $\delta$18O records from the different caves. However, I feel that we are left without clear conclusions or discussion of the implications when no firm conclusion can be drawn. This section needs to be written in a more concise and structured way (the authors should consider breaking the text into sub-sections), with a better highlight of the take-home messages. When reorganising the discussion, the authors could have in mind the following key questions to structure the text: (1) what is investigated and on what scientific ground? (2) what is observed? Is it significant or not? (3) What are the implications and how to go further?

- I find the Section 4.1 on the coherence and updates to NALPS19 versus NALPS unsatisfying. I believe that more specific justifications for selecting one speleothem rather than another to build the new composite calcite $\delta$18O record are missing. For instance, it would be useful to provide a quantitative comparison (in a table?) for at least one or two periods (if not all) where there is overlap between "old" and "new speleothems" to better illustrate that the new ones are better dated and hence, more appropriate than the ones already published to constitute the new composite curve.

My second general comment is that I find that many formulated statements, whether it is in the abstract or in the main manuscript, are too vague and/or miss some short background information. It renders the text sometimes hard to follow, especially for non-specialist readers. For instance, in multiple places, the authors state the good agreement of the different chronologies from the speleothem and ice core record within the dating uncertainties, without ever explicitly attached to their statement quantitative estimates of what those uncertainties are (pluri-decadal-scale? centennial-scale?). Another example is the lack of a short sentence providing basic information regarding the different ice core timescales discussed in the text. In the section 2 of my review,

I point out specific places in the manuscript that require revisions. But the authors should go through the manuscript with this comment in mind and revise accordingly when appropriate. I detail below some specific comments and technical corrections that should be considered by the authors when preparing the revised version.

2- SPECIFIC COMMENTS:

- While it is relatively long, I find that several statements in the abstract are too vague and should be reformulated to be more specific:

Line 14: ". . .with highly similar shifts": this is vague, spell out clearly that you are referring to abrupt changes observed in the water isotopic profiles from Greenland ice core. I think also that one should be careful with the use of "highly similar", they are not the same proxy. If such comparison is kept, it should be specified in which sense they are highly similar.

Line 18: It is necessary to specify in which term(s) the major transitional events between stadials and interstadials agree i.e. timing of the transitions and/or amplitude of the transitions? In the same sentence, it is necessary to provide also a quantitative average estimate of the uncertainties that are referred to here.

Line 19: ". . .a good agreement between the NALPS19 speleothem $\delta$18O record, the GICC05modelext NGRIP ice-core $\delta$18O record and. . .." First, my comment is the same as previously, it is important to make it clear in which term the good agreement is. Second, "GICC05modelext NGRIP ice core $\delta$18O record" should be reformulated. It needs to be clearer here that GICC05modelext refers to an ice core age model (it might not be necessarily obvious to all CP readers). It could be reformulated such as "the NGRIP ice core $\delta$18O record displayed on the GICC05modelext age scale".

Line 21: ". . .too young" and ". . .a longer duration". By how much? Please be quantitative here and provide at least an order of magnitude.

- For clarity purposes, I think it is important that throughout the manuscript, the authors specify "calcite δ18O" when mentioning the δ18O records from the different speleothems and "ice δ18O" and referring to the δ18O from ice cores. They do it in places, but I think this should appear systematically to avoid any confusion.

- P2, line 40: While for further details, the reader can certainly be referred to the Erhardt et al. (2019), a few sentences need to be added to describe the added value of performing such analysis and the general principle and method used for the ramp-fitting of the transitions.

- From P5, line 25: "Results" section:

SI Table 1 could appear in the main manuscript and information could be removed from the section. I found the information provided in the text very technical and from a non-speleothem expert view, I feel that this should better be the supplementary material. Instead the results section could be focused on the description of the different records and a detailed comparison of the timing inferred for the transitions in the paleoclimatic records using the statistical tool of Erhardt et al. (2019).

The authors discuss the relationship between calcite δ18O and calcite δ13C and perform Hendy test. Again, from a non-expert point of view, I would find it very useful to have a few of sentences explaining why they are performing such exercise, what they expect to be able to decipher from such investigation and finally what are the implications of the results of their test.

- P8, line 13: sentence starting with "Furthermore, despite...". Is there any explanation why the St Beatus records would record a signal that is different from the Gassel samples? It would be useful to provide more information on this.

- P9, line 5: The paragraph regarding the durations of GS-22 and the precursor event is difficult to follow. It would be very helpful if the authors could provide a table that summarises the different existing and new estimates of the durations of GS-22, GS-21.2 and GI-21.2 in the discussed paleoclimatic records. Implications from their new

NALPS composite curve should be expressed more explicitly.

- The numbering of the different sections needs to be revised. There should not be a sub Section 1.1 if there is no Section 1.2 within the introduction section. I have a similar comment with the sub section 3.2.1, 3.3.1 and 3.3.4.

3- FIGURES

- Figure 3.c. A sentence to explain the shift in calcite δ18O values between the Asian monsoon composite records and the original data from which it was constructed need to be added.

- Figure 3. In the last sentence of the caption, "NGRIP nomenclature" should be replaced by "the latest INTIMATE event stratigraphy scheme".

- Figure 4. This figure needs to be reworked to improve its readability. A y axis scale is missing. Transitions should be numbered following the INTIMATE event stratigraphy scheme and it should probably also show the reference curves in the background the reference curves onto which they have performed the analysis. Also, it needs to be clarified what are the three panels in (c), which speleothem records have been used to perform the transition analyses. Again, this would be straightforward if the original curves were shown underneath or in parallel.

- Figure 6. This figure is hard to read. Efforts must be made to improve its clarity. For instance, a triangle symbol should not be used to represent different parameters e.g. in (a), the catchment elevation relative to longitude and in the other panels some stadial δ18O values. For panels (e) and (g), "specific time periods" is vague, they should be specified. As far as I understand the caption for panel (f) is incorrect as only the mean δ18O values for the speleothems covering some selected GI and GS are being shown relative to the catchment elevation and not all. Finally, the expression "Colours are the same as in (a)" doesn't need to appear after the description of every panel. The authors could simply write the colour code at the start of the caption stating that it is

the same on all the panels of the figure.

- Figure 7. The authors should be explicit on which type of $\delta$18O values they are showing on the title of the axis e.g. ice $\delta$18O for (a), calcite $\delta$18O for (b) benthic $\delta$13C and planktic $\delta$18O.

- Figure S7. More information must be given to understand clearly what is represented: Titles for the two y axes should be provided as well as a description in the caption of the different curves that are represented e.g. $\delta$18O data, uncertainty ranges, probability density plots about the onset, mid-point and end of transition etc.

4- STYLISTIC, TYPOGRAPHICAL COMMENTS AND MINOR COMMENTS

P1, line 16: a space is missing between "using," and "eleven ".

P1, line 21: Since it is not mentioned previously, it is important here to specify that AICC2012 refers to an ice core chronology i.e. "NGRIP ice $\delta$18O when displayed on the AICC2012 ice core chronology ", or alternatively, the acronym can be spelt out.

P1, line 28: "precursor" instead of "pre-cursor".

P1, line 29: to write "GS-24.2 COOLING event".

P1, line 29: "...occurred shortly". Please be more specific so we have an idea from the abstract if you are talking about a few decades, or a few centuries, etc.

P1, line 35: write "orbital-".

P2, line 2: "...have been shown to be synchronous within dating uncertainties", please provide a reference to support this statement.

P2, line 7: There is no need for higher resolved ice $\delta$18O profile to identify the decadal- and centennial-scale variability, it was already visible from the $\delta$18O profile published in NGRIP project members 2004. Only that no one had provided a specific description before the study by Capron et al. (2010). Hence, I think that the sentence should be

rephrased.

P2, line 9: write "centennial-".

P2, line 14: "GICC05...." Add information regarding the time interval covered by each of the timescales.

P2, line 32: the authors are correct about the age differences between the different chronologies and they should provide a quantitative estimate of them (at least an order of magnitude).

P2, line 35: add a space between (ka) and (Boch et al. 2011).

P2, line 42: "a good agreement". Please be quantitative here regarding the agreement.

P3, line 3: why 1.1 Regional climate while there is no 1.2 and it follows the long introduction that doesn't have a sub-section heading.

P3, lines 17, 20 and 21: Northern Alps and Southern Alps.

P3, line 23: The formulation is awkward and should be rephrased with a more direct style.

P3, line 32: space is missing between (2015) and (though.

P5, line 26: for clarity purposes, please write instead "samples from Baschg Cave" and similarly in the titles of sub-sections 3.2, 3.3, 3.4 and 3.5.

P8, line 29: here and throughout the manuscript: Erhardt et al. 2019 (not 2018).

P9, line 2: In the paper by Columbu et al. (2017), a well-dated Sardinian speleothem covering GI-25b and GI-25a is presented. The timing of the abrupt transitions is also discussed and compared relative to the timing of the same events when displayed on the different Greenland ice core timescales. This study also provides evidences that there is a good agreement between the transition timing in the speleothem record and when considering GICC05modelext timescale, but that when considering

the AICC2012 chronology, ages are younger by several millennia. The authors should mention this study in their manuscript.

P9, line 3: The sentence should be completed: "...too young by about XX yrs".

P9, line 7: The formulation of the sentence starting by "This demonstrated...." is awkward. It needs to be reformulated.

P9, line 13: I don't find the information in brackets necessary, it can probably be removed.

P9, line 22: I find the title of the section 4.3 quite vague and not really appropriate. The authors should try and be more specific.

P12, line 11: centennial-

P12, line 14: space between (Fig. 7) and (Capron...).

P12, lines 13 and 27 and P13, lines 17, 18, 20 and 22: The use of the word "termination" should be avoided in this context and replaced by e.g stadial-interstadial transition. Indeed, as the authors know the word "termination" is classically used in paleoclimatology to refer to glacial-interglacial transitions and I think for clarity purposes, it is preferable to avoid introducing this term in a different context and to refer to a different climatic event.

P12, line 19: "Changes IN Ca2+" rather than "Changes TO Ca2+". Also, I don't find the reference to Rasmussen et al. (2014) appropriate in this context. Instead I would suggest referring to the studies by Ruth et al. (2007). U. Ruth et al., Geophys. Res. Lett. 34, L03706, 10.1029/2006GL027876 (2007).

P12, line 25: "the NGRIP nomenclature" should be replaced by "the INTIMATE event stratigraphy scheme".

P13, line 13: space between (Wang et al. 2004) and (Fig. 7).

REFERENCES:

Boch et al. Clim. Past, 7, 1247-1259, https://doi.org/10.5194/cp-7-1247-2011, 2011.

Capron et al. Clim. Past, 6, 345-365, https://doi.org/10.5194/cp-6-345-2010, 2010.

Columbu et al. QSR, 169, 391-397, 2017.

Erhardt et al. Clim. Past, 15, 811-825, https://doi.org/10.5194/cp-15-811-2019, 2019.

Ruth et al., Geophys. Res. Lett. 34, L03706, 10.1029/2006GL027876, 2007.

Wolff et al. QSR, 29, 2828-2838, 2010.

---

## Referee Comment (RC2) · Anonymous Referee #2 · 4 Jun 2019

General comments: Moseley et al. present an updated version of the northern Alps speleothem d18O record for the last glacial (NALPS19). Earlier versions of this record have provided crucial information for the interpretation of the timing and expression of sub-orbital glacial climate change in central Western Europe and their close connection to Greenland. Moreover, given the possibility to obtain high-precision U-Th ages from stalagmites, these records have the potential to help refine crucial, but so far debated, portions of the Greenland ice core record chronology. With the new NALPS19 record, the authors significantly increase the temporal coverage of the record during the last glacial, and improve previous versions of the record using more precisely dated stalagmites. This allows them to compare the NALPS19 chronology with different Greenland

ice core chronologies, with the potential to refine the latter. This is for example the case for the duration of the period from GS-22 to GI-21.2, for which the authors propose a new duration based on the speleothem record. Moreover, the authors discuss variability and influencing processes on speleothem d18O in their record, and possible biases in some locations, although they acknowledge that these interpretations remain often preliminary and will require a more dense dataset, possibly coupled to model simulations. Finally, the authors also discuss the presence of centennial-scale "precursor" events, which are present in both Greenland ice core records and the NALPS19 speleothems, and which they interpret to be related to meltwater pulses accompanying stadial-interstadial transitions.

Overall, this study is of importance to the broader palaeoclimate community, as it provides a significant advance in the coverage of the last glacial in the Alpine region, and is therefore within the scope of Climate of the Past. The study is well-designed and provides new data that allows substantial conclusions as to the close connection between Greenland and northern Alps during the last glacial, as well as firm geochronological anchoring points for some previously debated Greenland stadial/interstadial events. The methods used are well-described and valid, and overall the results support the interpretations.

My main point of critique is in the sometimes lengthy discussion, which can be difficult to follow for non-experts in both ice cores and speleothem science. I therefore suggest the authors provide moderate revisions (see specific and technical comments) to the manuscript before it can be accepted for publication.

Specific comments: Discussion of chronology (section 4.2): This section is in parts difficult to follow, especially for readers not overly familiar with the ice core literature. Some studies are cited, but the reasoning for this is not explained, and this can be confusing. I suggested some instances where some more background would be beneficial to improve overall clarity (see technical comments).

This is especially the case for the discussion of GS-22. I think it would be worthwhile to restructure this paragraph and clarify the main message, i.e., NALPS19 allows to re-evaluate conflicting results from different ice core age modelling techniques, and this is especially clear for the interval between GS-22-GS-21.2.

Discussion of palaeoclimate and d18O (section 4.3): I am a bit confused with the treatment of the Siebenhengste record. At the beginning of the section, the authors exclude the LGM part of the record from Siebenhengste from their discussion on the range in d18O, because of the influence from different moisture sources previously inferred for this time period. Here I was hoping the authors could provide some more background as to why this moisture source effect is only seen in the 7H LGM record: is it due to the time period covered or is the location of the cave the likely reason for this? Why are the authors certain that changes in moisture source were not an issue for any of the other records in the compilation? Further along in the text, there is a lengthy discussion of why the Siebenhengste record is anomalous, but there is no more mention if the source effect. I think it would greatly benefit the flow of the manuscript if the authors could elaborate a bit more on their reasoning for this, and link it back to the beginning of the section and the discussion on source changes during the LGM.

Discussion of stadial-level centennial-scale cold events (section 4.4): I think these events need to be more clearly pointed out in the figure 7, or even in a separate, zoomed-in figure, as it is not particularly clear what is meant now.

Technical comments: Page 1: - line 21: The meaning of AICC2012 needs to be specified, otherwise this sentence is very confusing for non-experts. - line 37: please add "oxygen" to "isotopic records" to clarify what is meant.

Page 2: - lines 31 and following: I think here the authors must clearly state that this chronological issue is also present in the ice cores and not only between the NALPS record and the ice cores.

Page 3: - line 2: "controlling the d18O of precipitation in this region" would be more precise. - line 20: "the northern Alps receive" (instead of receives)

Page 4: The description of the sites and stalagmites is a bit lengthy and confusing in parts. I wonder if it would be better for the flow of the manuscript to summarise these details in a table, and streamline the text?

Page 5: Lines 28 and following throughout the sample description: U concentrations should be in ng/g (not ug/g) according to Table 2.

Page 6: For the caves with only one stalagmite analysed, it would be better to merge the two headings into one.

Page 8: Lines 27 and following: please add some context here regarding the chronologies GICC05modelext and AICC2012, otherwise it is difficult to follow for readers less familiar with ice cores.

Page 9: Lines 2-4: Please provide a brief explanation of what the findings of Extier are for the readers not familiar with this study.

Line 5: Please add the ages of the GS-22 interval here for context. Also, given that this is discussed at length over the next section, I would appreciate if the authors could point out this interval (and GI-GS21.2) in Fig. 3 or 5.

Lines 6-7: "Vallelonga et al. (2012) proposed the duration of GS-22 to be 2,894 $\pm$ 198 years and GI-21.2 - GS-21.2 to be 350 $\pm$ 19 years (together 3,244 $\pm$ 199 years, two sigma error)." This sentence reads confusingly to me: I assume the authors mean that the duration of the transition between GI-21.2 and GS-21.2 to be 350 years, while the entirety of the interval is 3244 years?

Line 9: NGRIP-EDML should be explained.

Page 10: Line 10: "The highest and lowest $\delta$18O values for stadials and interstadials also both come from the same caves." I find this sentence confusing: the highest and

lowest in general? Which cave are these values from?

Line 27: I would rephrase "mean d18O" to "mean d18O of an entire record" or similar.

Line 35: "For a given location, however, Ambach et al. (1968) argued that the altitude effect cannot be the result of a difference in condensation temperature, because the condensation level should be approximately the same." I find this sentence confusing, and would also appreciate some more details on why the condensation level is the same.

Page 13: Line 32-35: "Furthermore, we suggest that the highly-debated GS-22 - GI-21.2 - GS-21.2 interval had a duration of 3,857 $\pm$ 249 years, which is in closer agreement with the 4,121 $\pm$ 325 years of NGRIP-EDML (Capron et al., 2010b) and the 3,793 $\pm$ 805 years of the Asian monsoon composite (Kelly et al.,2006; 35 Kelly, 2010; Cheng et al., 2016)." Closer agreement than what?

Figures: Figure 3: In the caption, I believe there is some information missing. For c) only the Dongge data is referenced, and there is no mention of Hulu. There is also a repetition at f) "for (e) colour-coded the same". Figure 4: I think this figure would benefit from some additional work. For example, it would be clearer if the different ice cores (b) and stalagmites (c) for which the ramp-fitting was done were indicated in the figure with labels. Also, possibly adding labels for the transitions identified in the Greenland records would help. Figure 5: Here it would be helpful to the reader if the records were labelled, as in figure 2. Figure 6: I think it would be helpful to have a legend in the figure showing which symbol belongs to which cave.

---

## Author Comment (AC1) · 29 Jul 2019

Moseley et al. would like to thank both reviewers for their insightful comments that provide excellent guidance for improving the manuscript. In summary, we agree with the majority of the comments and are happy to revise the manuscript accordingly.

**Response to Reviewer 1**

R1 comment: Section 2.1 presenting the cave sites and speleothems could be shortened. Instead of a long and tedious text to read, it would be helpful to have an associated table in the main manuscript that summarizes at least some of the information provided in the text e.g. cave name, coordinates, elevation, air temperature and precipitation rate, associated sample acronyms, length of the samples...).

**AC: this is a helpful suggestion, which we agree with and are prepared to revise the manuscript accordingly.**

R1 comment: The discussion is also difficult to follow in Section 4.3, and the take-home messages hard to identify. The authors are investigating and discussing the roles of the different potential control factors on the calcite  $\delta^{18}$ O records from the different caves. However, I feel that we are left without clear conclusions or discussion of the implications when no firm conclusion can be drawn. This section needs to be written in a more concise and structured way (the authors should consider breaking the text into sub-sections), with a better highlight of the take-home messages. When reorganising the discussion, the authors could have in mind the following key questions to structure the text: (1) what is investigated and on what scientific ground? (2) what is observed? Is it significant or not? (3) What are the implications and how to go further?

AC: indeed, it is clearly stated that "We appreciate that our investigation is only a first approximation, however, a more thorough investigation would require modelling which is beyond the scope of this study." Hence, firm conclusions are not drawn. This is the first time that we have a near complete view of the Northern Alps d18O calcite record, hence it is an opportunity to take a first look at the controlling factors. We agree this is long-winded, and will add a summary at the end of this section with the take-home message, which is that in the majority of cases, the d18O values are as we would expect – i.e. d18O becomes more depleted with distance from source (continental effect), and more depleted with elevation (altitude effect). We do, however, find that there are some anomalies to this, which seem to affect the very high altitude caves only – at times these are less depleted than would be expected.

R1 comment: I find the Section 4.1 on the coherence and updates to NALPS19 versus NALPS unsatisfying. I believe that more specific justifications for selecting one speleothem rather than another to build the new composite calcite  $\delta$ 180 record are missing. For instance, it would be useful to provide a quantitative comparison (in a table?) for at least one or two periods (if not all) where there is overlap between "old" and "new speleothems" to better illustrate that the new ones are better dated and hence, more appropriate than the ones already published to constitute the new composite curve.

AC: for the most part, this is not a composite curve, e.g. the transition between Gl24.1 and GS24.1 includes GAS22, GAS29 and HUN14, whilst the transition from Gl21.1 to GS21.1 includes GAS12, GAS13 and SCH6 for the very reason that it is not possible to say that one speleothem chronology is better than another. The overlap between old and new speleothems is already provided as SI Fig 5. There it is possible to see the coherence between the new samples, and also to see that the precision of some dating is better than others (e.g. GAS 25 versus BA1 or BA7). The other information to draw on is that BA samples are slightly dirty, whereas GAS samples are not. We

**mention that we have removed the 'dirty' samples where cleaner ones are available but are not explicit about which ones – we will change this accordingly.**

R1 comment: My second general comment is that I find that many formulated statements, whether it is in the abstract or in the main manuscript, are too vague and/or miss some short background information. It renders the text sometimes hard to follow, especially for non-specialist readers. For instance, in multiple places, the authors state the good agreement of the different chronologies from the speleothem and ice core record within the dating uncertainties, without ever explicitly attached to their statement quantitative estimates of what those uncertainties are (pluri-decadal-scale? centennial-scale?).

AC: if the ages of specific events are the same within dating uncertainties, then to make statements about specific decadal, centennial timescales etc. would be to ignore those uncertainties. A better phrase maybe to say that 'events occur synchronously within dating uncertainties' rather than they are 'in good agreement within uncertainty'.

Another example is the lack of a short sentence providing basic information regarding the different ice core timescales discussed in the text. In the section 2 of my review, I point out specific places in the manuscript that require revisions. But the authors should go through the manuscript with this comment in mind and revise accordingly when appropriate. I detail below some specific comments and technical corrections that should be considered by the authors when preparing the revised version.

**AC: this will be revised according to the suggestions later in the review.**

R1 comment: Line 14: "...with highly similar shifts": this is vague, spell out clearly that you are referring to abrupt changes observed in the water isotopic profiles from Greenland ice core. I think also that one should be careful with the use of "highly similar", they are not the same proxy. If such comparison is kept, it should be specified in which sense they are highly similar.

**AC: Agreed, this will be revised accordingly.**

R1 comment: Line 18: It is necessary to specify in which term(s) the major transitional events between stadials and interstadials agree i.e. timing of the transitions and/or amplitude of the transitions? In the same sentence, it is necessary to provide also a quantitative average estimate of the uncertainties that are referred to here.

**AC: Agreed, this will be revised accordingly.**

R1 comment: Line 19: "...a good agreement between the NALPS19 speleothem  $\delta$ 180 record, the GICC05modelextNGRIP ice-core  $\delta$ 180 record and...." First, my comment is the same as previously, it is important to make it clear in which term the good agreement is.

**AC: See previous comment.**

Second, "GICC05modelext NGRIP ice core  $\delta$ 18O record" should be reformulated. It needs to be clearer here that GICC05modelext refers to an ice core age model (it might not be necessarily obvious to all CP readers). It could be reformulated such as "the NGRIP ice core  $\delta$ 18O record displayed on the GICC05modelext age scale".

AC: this seems very wordy but is clearer, it will be revised accordingly.

Line 21: "...too young" and "...a longer duration". By how much? Please be quantitative here and provide at least an order of magnitude.

**AC: Agreed, this will be revised accordingly.**

R1 comment: For clarity purposes, I think it is important that throughout the manuscript, the authors specify "calcite  $\delta$ 18O" when mentioning the  $\delta$ 18O records from the different speleothems and "ice  $\delta$ 18O" and referring to the  $\delta$ 18O from ice cores. They do it in places, but I think this should appear systematically to avoid any confusion.

**AC: Agreed, this will be revised accordingly.**

R1 comment: P2, line 40: While for further details, the reader can certainly be referred to the Erhardt et al. (2019), a few sentences need to be added to describe the added value of performing such analysis and the general principle and method used for the ramp-fitting of the transitions.

**AC: Agreed, this will be revised accordingly.**

R1 comment: SI Table1 could appear in the main manuscript and information could be removed from the section. AC: we consider such information to be 'extra' and not essential to understanding the manuscript; however, we can add more to the main text if desired. I found the information provided in the text very technical and from a non speleothem expert view, I feel that this should better be the supplementary material. AC: Agreed, this will be revised accordingly and produce a table. Instead the results section could be focused on the description of the different records and a detailed comparison of the timing inferred for the transitions in the paleoclimatic records using the statistical tool of Erhardt et al. (2019).

R1 comment: The authors discuss the relationship between calcite  $\delta$ 18O and calcite  $\delta$ 13C and perform Hendy test. Again, from a non-expert point of view, I would find it very useful to have a few of sentences explaining why they are performing such exercise, what they expect to be able to decipher from such investigation and finally what are the implications of the results of their test.

**AC: Agreed, this will be revised accordingly.**

R1 comment: P8, line 13: sentence starting with "Furthermore, despite...". Is there any explanation why the St Beatus records would record a signal that is different from the Gassel samples? It would be useful to provide more information on this.

**AC: We do not know why some of the bigger-scale climate record appears to be missing from St Beatus, but we can add a sentence or two to explain this.**

R1 comment: P9, line 5: The paragraph regarding the durations of GS-22 and the precursor event is difficult to follow. It would be very helpful if the authors could provide a table that summarises the different existing and new estimates of the durations of GS-22, GS21.2 and GI-21.2 in the discussed paleoclimatic records. Implications from their new NALPS composite curve should be expressed more explicitly.

**AC: Agreed, this will be revised accordingly.**

R1 comment: The numbering of the different sections needs to be revised. There should not be a subsection 1.1 if there is no Section 1.2 within the introduction section. I have a similar comment with the sub section 3.2.1, 3.3.1 and 3.3.4.

AC: This is not logical. The introduction didn't require a third independent section, hence there is no section 1.2. The results section is split accordingly: X Result. X.X Caves X.X.X. Samples. It would not make sense for specific samples to be a high level designation.

R1 comment: Figure 3.c. A sentence to explain the shift in calcite  $\delta$ 180 values between the Asian monsoon composite records and the original data from which it was constructed need to be added.

AC: We consider this beyond the scope of this manuscript since this shift is a product of the work by the original authors who compiled the Asian monsoon composite. If the reader wishes to know further details, they should refer to the original work.

R1 comment: Figure 3. In the last sentence of the caption, "NGRIP nomenclature" should be replaced by "the latest INTIMATE event stratigraphy scheme".

**AC: Agreed, this will be revised accordingly.**

R1 comment: Figure 4. This figure needs to be reworked to improve its readability. A y axis scale is missing. Transitions should be numbered following the INTIMATE event stratigraphy scheme and it should probably also show the reference curves in the background the reference curves onto which they have performed the analysis. Also, it needs to be clarified what are the three panels in (c), which speleothem records have been used to perform the transition analyses. Again, this would be straightforward if the original curves were shown underneath or in parallel.

**AC: Agreed, this will be revised accordingly.**

R1 comment: Figure 6. This figure is hard to read. Efforts must be made to improve its clarity. For instance, a triangle symbol should not be used to represent different parameter se.g. in (a), the catchment elevation relative to longitude and in the other panels some stadial  $\delta$ 180 values. For panels (e) and (g), "specific time periods" is vague, they should be specified. As far as I understand the caption for panel (f) is incorrect as only the mean  $\delta$ 180 values for the speleothems covering some selected GI and GS are being shown relative to the catchment elevation and not all. Finally, the expression "Colours are the same as in (a)" doesn't need to appear after the description of every panel. The authors could simply write the colour code at the start of the caption stating that it is the same on all the panels of the figure.

**AC: Agreed, this will be revised accordingly.**

R1 comment: Figure 7. The authors should be explicit on which type of  $\delta$ 180 values they are showing on the title of the axis e.g. ice  $\delta$ 180 for (a), calcite  $\delta$ 180 for (b) benthic  $\delta$ 13C and planktic  $\delta$ 180.

**AC: Agreed, this will be revised accordingly.**

R1 comment: FigureS7. More information must be given to understand clearly what is represented: Titles for the two y axes should be provided as well as a description in the caption of the different curves that are represented e.g.  $\delta$ 18Odata,uncertainty ranges,probability density plots about the onset, mid-point and end of transition etc.

**AC: Agreed, this will be revised accordingly.**

R1 comment: P1, line 16: a space is missing between "using," and "eleven ".

AC: Agreed, this will be revised accordingly.

R1 comment: P1, line 21: Since it is not mentioned previously, it is important here to specify that AICC2012 refers to an ice core chronology i.e. "NGRIP ice  $\delta$ 180 when displayed on the AICC2012 ice core chronology ", or alternatively, the acronym can be spelt out.

**AC: Agreed, this will be revised accordingly.**

R1 comment: P1, line 28: "precursor" instead of "pre-cursor".

AC: Agreed, this will be revised accordingly.

R1 comment: P1, line 29: to write "GS-24.2 COOLING event".

AC: Agreed, this will be revised accordingly.

R1 comment: P1, line 29: "...occurred shortly". Please be more specific so we have an idea from the abstract if you are talking about a few decades, or a few centuries, etc.

AC: Agreed, this will be revised accordingly.

R1 comment: P1, line 35: write "orbital-".

AC: Agreed, this will be revised accordingly.

P2, line 2: "...have been shown to be synchronous within dating uncertainties", please provide a reference to support this statement.

**AC: The reference is already there, it is at the end of the sentence.**

R1 comment: P2, line 7: There is no need for higher resolved ice  $\delta$ 180 profile to identify the decadal and centennial-scale variability, it was already visible from the  $\delta$ 180 profile published in NGRIP project members 2004. Only that no one had provided a specific description before the study by Capron et al. (2010). Hence, I think that the sentence should be rephrased.

AC: Agreed, this will be revised accordingly.

R1 comment: P2, line 9: write "centennial-".

AC: Agreed, this will be revised accordingly.

R1 comment: P2, line 14: "GICC05...." Add information regarding the time interval covered by each of the timescales.

AC: Agreed, this will be revised accordingly.

R1 comment: P2, line 32: the authors are correct about the age differences between the different chronologies and they should provide a quantitative estimate of them (at least an order of magnitude).

AC: Agreed, this will be revised accordingly.

R1 comment: P2, line 35: add a space between (ka) and (Boch et al. 2011).

AC: Agreed, this will be revised accordingly.

R1 comment: P2, line42: "a good agreement". Please be quantitative here regarding the agreement.

AC: This is difficult to comprehend. Since they are in agreement with respect to timing, how should one be quantitative unless the uncertainties are ignored? It would only be possible to be quantitative if they were out of agreement or the uncertainties are not regarded.

R1 comment: P3, line 3: why 1.1 Regional climate while there is no 1.2 and it follows the long introduction that doesn't have a sub-section heading.

AC: There is no 1.2 because the next topic is methods and therefore requires the beginning of a new chapter. We consider Regional Climate to be Introductory Material, but could also consider it placing it as its own section 2. Regional Climate

R1 comment: P3, lines 17, 20 and 21: Northern Alps and Southern Alps.

AC: Agreed, this will be revised accordingly.

R1 comment: P3, line 23: The formulation is awkward and should be rephrased with a more direct style.

AC: Agreed, this will be revised accordingly.

R1 comment: P3, line 32: space is missing between (2015) and (though.

AC: Agreed, this will be revised accordingly.

R1 comment: P5, line 26: for clarity purposes, please write instead "samples from Baschg Cave" and similarly in the titles of sub-sections 3.2, 3.3, 3.4 and 3.5.

AC: Agreed, this will be revised accordingly.

P8, line 29: here and throughout the manuscript: Erhardt et al. 2019 (not 2018).

AC: Agreed, this will be revised accordingly.

R1 comment: P9, line 2: In the paper by Columbu et al. (2017), a well-dated Sardinian speleothem covering GI-25b and GI-25a is presented. The timing of the abrupt transitions is also discussed and compared relative to the timing of the same events when displayed on the different Greenland ice core timescales. This study also provides evidences that there is a good agreement between the transition timing in the speleothem record and when considering GICC05modelext timescale, but that when considering the AICC2012 chronology, ages are younger by several millennia. The authors should mention this study in their manuscript.

AC: Agreed, this will be revised accordingly.

R1 comment: P9, line 3: The sentence should be completed: "...too young by about XX yrs".

AC: Agreed, this will be revised accordingly.

R1 comment: P9, line 7: The formulation of the sentence starting by "This demonstrated...." is awkward. It needs to be reformulated.

AC: Agreed, this will be revised accordingly.

R1 comment: P9, line 13: I don't find the information in brackets necessary, it can probably be removed.

AC: We disagree, it is important to be explicit about what the datum is, otherwise these ages spread inaccurately throughout the literature.

R1 comment: P9, line 22: I find the title of the section 4.3 quite vague and not really appropriate. The authors should try and be more specific.

**AC: Agreed, this will be revised accordingly.**

R1 comment: P12, line 11: centennialP12, line 14: space between (Fig. 7) and (Capron...).

**AC: Agreed, this will be revised accordingly.**

R1 comment: P12, lines 13 and 27 and P13, lines 17, 18, 20 and 22: The use of the word "termination" should be avoided in this context and replaced by e.g stadial-interstadial transition. Indeed, as the authors know the word "termination" is classically used in paleoclimatology to refer to glacial-interglacial transitions and I think for clarity purposes, it is preferable to avoid introducing this term in a different context and to refer to a different climatic event.

**AC: Agreed, this will be revised accordingly.**

R1 comment: P12, line 19: "Changes IN Ca2+" rather than "Changes TO Ca2+". Also, I don't find the reference to Rasmussen et al. (2014) appropriate in this context. Instead I would suggest referring to the studies by Ruth et al. (2007). U. Ruth et al., Geophys. Res. Lett. 34, L03706, 10.1029/2006GL027876 (2007).

**AC: Agreed, this will be revised accordingly.**

R1 comment: P12, line 25: "the NGRIP nomenclature" should be replaced by "the INTIMATE event stratigraphy scheme".

AC: Agreed, this will be revised accordingly.

R1 comment: P13, line 13: space between (Wang et al. 2004) and (Fig. 7).

AC: Agreed, this will be revised accordingly.

**R1 comment: References**

AC: Agreed, this will be revised accordingly.

---

## Author Comment (AC2) · 29 Jul 2019

Moseley et al. would like to thank both reviewers for their insightful comments that provide excellent guidance for improving the manuscript. In summary, we agree with the majority of the comments and are happy to revise the manuscript accordingly.

**Response to Reviewer 2**

My main point of critique is in the sometimes lengthy discussion, which can be difficult to follow for non-experts in both ice cores and speleothem science. I therefore suggest the authors provide moderate revisions (see specific and technical comments) to the manuscript before it can be accepted for publication.

AC: Reviewer 1 made similar comments, hence the text will be revised accordingly.

Discussion of chronology (section 4.2): This section is in parts difficult to follow, especially for readers not overly familiar with the ice core literature. Some studies are cited, but the reasoning for this is not explained, and this can be confusing. I suggested some instances where some more background would be beneficial to improve overall clarity (see technical comments).

This is especially the case for the discussion of GS-22. I think it would be worthwhile to restructure this paragraph and clarify the main message, i.e., NALPS19 allows to reevaluate conflicting results from different ice core age modelling techniques, and this is especially clear for the interval between GS-22-GS-21.2.

AC: Agreed, this will be revised accordingly.

Discussion of palaeoclimate and d18O (section 4.3): I am a bit confused with the treatment of the Siebenhengste record. At the beginning of the section, the authors exclude the LGM part of the record from Siebenhengste from their discussion on the range in d18O, because of the influence from different moisture sources previously inferred for this time period.

AC: This is incorrect. It is excluded on the basis of different transport pathways. Moisture source alone would be a different concept.

Here I was hoping the authors could provide some more background as to why this moisture source effect is only seen in the 7H LGM record: is it due to the time period covered or is the location of the cave the likely reason for this?

AC: This is beyond the scope of this study. The LGM record is a paper already published and thus an in-depth discussion already belongs to the original study.

Why are the authors certain that changes in moisture source were not an issue for any of the other records in the compilation?

AC: Again, it is not a matter of moisture source but transport pathway. The reviewer is referred to the original paper for an explanation, but in short, the Siebenhengste record only records changes in transport pathway and associated Rossby wave breaking at the LGM and not for the remainder of the glacial. We therefore assume the same for the other records, and since we have none covering the LGM, this is not a problem for us.

Further along in the text, there is a lengthy discussion of why the Siebenhengste record is anomalous, but there is no more mention if the source effect. I think it would greatly benefit the flow of the manuscript if the authors could elaborate a bit more on their reasoning for this, and link it back to the beginning of the section and the discussion on source changes during the LGM.

AC: As already mentioned, the changes at the LGM are specific to this time period. The ice sheets were not big enough in MIS 5 to cause the same changes to transport pathways as at the LGM. Even in early MIS 2, such changes were not apparent at Siebenhengste.

Discussion of stadial-level centennial-scale cold events (section 4.4): I think these events need to be more clearly pointed out in the figure 7, or even in a separate, zoomed-in figure, as it is not particularly clear what is meant now.

AC: A zommed-in figure can be added to the SI

Page 1: - line 21: The meaning of AICC2012 needs to be specified, otherwise this sentence is very confusing for non-experts.

AC: Agreed, this will be revised accordingly.

- line 37: please add "oxygen" to "isotopic records" to clarify what is meant.

AC: Agreed, this will be revised accordingly.

Page 2: - lines 31 and following: I think here the authors must clearly state that this chronological issue is also present in the ice cores and not only between the NALPS record and the ice cores.

AC: we do not mention it is between NALPS and ice cores, just that it is present in both chronologies

Page 3: - line 2: "controlling the d18O of precipitation in this region" would be more precise. –

AC: Agreed, this will be revised accordingly.

- line 20: "the northern Alps receive" (instead of receives)

AC: Agreed, this will be revised accordingly.

Page 4: The description of the sites and stalagmites is a bit lengthy and confusing in parts. I wonder if it would be better for the flow of the manuscript to summarise these details in a table, and streamline the text?

AC: Agreed, this will be revised accordingly as per R1 also.

Page 5: Lines 28 and following throughout the sample description: U concentrations should be in ng/g (not ug/g) according to Table 2.

AC: Agreed, this will be revised accordingly.

Page 6: For the caves with only one stalagmite analysed, it would be better to merge the two headings into one.

AC: Agreed, this will be revised accordingly.

Page 8: Lines 27 and following: please add some context here regarding the chronologies GICC05modelext and AICC2012, otherwise it is difficult to follow for readers less familiar with ice cores.

AC: It is unclear what is meant by 'context'. A review of ice core chronologies is beyond the scope of this manuscript.

Page 9: Lines 2-4: Please provide a brief explanation of what the findings of Extier are for the readers not familiar with this study.

AC: Agreed, this will be revised accordingly.

Line 5: Please add the ages of the GS-22 interval here for context. Also, given that this is discussed at length over the next section, I would appreciate if the authors could point out this interval (and GI-GS21.2) in Fig. 3 or 5.

AC: Agreed, this will be revised accordingly.

Lines 6-7: "Vallelonga et al. (2012) proposed the duration of GS-22 to be 2,894±198 years and GI-21.2 - GS-21.2 to be 350 ± 19 years (together 3,244 ± 199 years, two sigma error)." This sentence reads confusingly to me: I assume the authors mean that the duration of the transition between GI-21.2 and GS-21.2 to be 350 years, while the entirety of the interval is 3244 years?

AC: Agreed, this will be revised accordingly.

Line 9: NGRIP-EDML should be explained.

AC: Agreed, this will be revised accordingly.

Page 10: Line 10: "The highest and lowest δ18O values for stadials and interstadials also both come from the same caves." I find this sentence confusing: the highest and lowest in general? Which cave are these values from?

AC: Agreed, this will be revised accordingly.

Line 27: I would rephrase "mean d18O" to "mean d18O of an entire record" or similar.

AC: Agreed, this will be revised accordingly.

Line 35: "For a given location, however, Ambach et al. (1968) argued that the altitude effect cannot be the result of a difference in condensation temperature, because the condensation level should be approximately the same." I find this sentence confusing, and would also appreciate some more details on why the condensation level is the same.

AC: The sentence clearly starts with 'for a given location', therefore if it rains at a given location but at different altitudes, then the condensation occurred at the same point, because it is the same rain event.

Page 13: Line 32-35: "Furthermore, we suggest that the highly-debated GS-22 - GI21.2 - GS-21.2 interval had a duration of 3,857 ± 249 years, which is in closer agreement with the 4,121±325 years of NGRIP-EDML (Capron et al., 2010b) and the 3,793 ±805 years of the Asian monsoon composite (Kelly et al.,2006; 35 Kelly, 2010; Cheng et al., 2016)." Closer agreement than what?

AC: Agreed, this will be revised accordingly.

Figures: Figure 3: In the caption, I believe there is some information missing. For c) only the Dongge data is referenced, and there is no mention of Hulu. AC: This Hulu data is part of the Cheng et al 2016, but it could be more explicit. There is also a repetition at f) "for(e)colour-coded the same". AC: Agreed, this will be revised accordingly.

Figure4: I think this figure would benefit from some additional work. For example, it would be clearer if the different ice cores (b) and stalagmites (c) for which the ramp-fitting was done were

indicated in the figure with labels. Also, possibly adding labels for the transitions identified in the Greenland records would help. AC: Agreed, this will be revised accordingly as per R1 also.

Figure 5: Here it would be helpful to the reader if the records were labelled, as in figure 2. AC: Agreed, this will be revised accordingly.

Figure 6: I think it would be helpful to have a legend in the figure showing which symbol belongs to which cave.

AC: Agreed, this will be revised accordingly.

---

## Author Response (AR1)

University of Innsbruck
Institute of Geology
Innrain 52
Innsbruck 6020
Austria

10$^{th}$ September, 2019

Dear Dominik,

I would like to submit the revised version of the manuscript entitled "NALPS19: Sub-orbital scale climate variability recorded in Northern Alpine speleothems during the last glacial period" for re-evaluation in Climate of the Past. The comments from both of the reviewers were extremely helpful and justified and have helped greatly to improve the manuscript. Each of the comments are addressed in turn below.

**General Comments of Reviewer 1**

- Section 2.1 presenting the cave sites and speleothems could be shortened. Instead of a long and tedious text to read, it would be helpful to have an associated table in the main manuscript that summarizes at least some of the information provided in the text e.g. cave name, coordinates, elevation, air temperature and precipitation rate, associated sample acronyms, length of the samples...).

AC: This has been done. The information can now be found in table 1. Additionally we shortened the lengthy U-Th dating results section and put this information in a new summary table 2.

- The discussion is also difficult to follow in Section 4.3, and the take-home messages hard to identify. The authors are investigating and discussing the roles of the different potential control factors on the calcite δ18O records from the different caves. However, I feel that we are left without clear conclusions or discussion of the implications when no firm conclusion can be drawn. This section needs to be written in a more concise and structured way (the authors should consider breaking the text into sub-sections), with a better highlight of the take-home messages. When reorganising the discussion, he authors could have in mind the following key questions to structure the text: (1)what is investigated and on what scientific ground? (2) what is observed? Is it significant or not? (3) What are the implications and how to go further?

AC: This has been greatly simplified and a short summary section added to the end to highlight the take home message.

- I find the Section 4.1 on the coherence and updates to NALPS19 versus NALPS unsatisfying. I believe that more specific justifications for selecting one speleothem rather than another to build the new composite calcite δ18O record are missing. For instance, it would be useful to provide a quantitative comparison (in a table?) for at least one or two periods (if not all) where there is overlap between "old" and "new speleothems" to better illustrate that the new ones are better dated and hence, more appropriate than the ones already published to constitute the new composite curve.

AC: This has been done and we now give a step by step approach as to why some speleothems are included and others are excluded. A table was not possible for this, hence the comparison is in the text. One can also see the comparison of the records in SI Fig. 5 (which was always the case).

My second general comment is that I find that many formulated statements, whether it is in the abstract or in the main manuscript, are too vague and/or miss some short background information. It renders the text sometimes hard to follow, especially for non-specialist readers. For instance, in multiple places, the authors state the good agreementofthedifferentchronologiesfromthespeleothemandicecorerecordwithin the dating uncertainties, without ever explicitly attached to their statement quantitative estimates of what those uncertainties are (pluridecadal-scale? centennial-scale?). Another example is the lack of a short sentence providing basic information regarding the different ice core timescales discussed in the text. In the section 2 of my review, C3

AC: This has been altered now so that quantitative statements are used throughout the manuscript to support the statements.

**Specific Comments of Reviewer 1**

- While it is relatively long, I find that several statements in the abstract are too vague and should be reformulated to be more specific:

Line 14: "...with highly similar shifts": this is vague, spell out clearly that you are referring to abrupt changes observed in the water isotopic profiles from Greenland ice core. I think also that one should be careful with the use of "highly similar", they are not the same proxy. If such comparison is kept, it should be specified in which sense they are highly similar.

AC: this has been revised to 'enabling direct chronological comparisons', P1 L15-16

Line 18: It is necessary to specify in which term(s) the major transitional events between stadials and interstadials agree i.e. timing of the transitions and/or amplitude of the transitions? In the same sentence, it is necessary to provide also a quantitative average estimate of the uncertainties that are referred to here.

AC: This has been updated accordingly to ....."Where speleothems grew synchronously, the timing of major transitional events in $\delta^{18}O_{calc}$ between stadials and interstadials (and vice versa) are all in agreement on multi-decadal timescales." P1 L20-21

Line 19: "...a good agreement between the NALPS19 speleothem δ18O record, the GICC05modelextNGRIP ice-core δ18Orecord and...." First, my comment is the same as previously, it is important to make it clear in which term the good agreement is.

AC: This has been updated accordingly to ....."Ramp-fitting analysis further reveals that, with the exception of stadial-20, the timing of $\delta^{18}O$ transitions occurred synchronously within centennial-scale dating uncertainties between." P1 L 21-23

Second, "GICC05modelext NGRIP ice core δ18O record" should be reformulated. It needs to be clearer here that GICC05modelext refers to an ice core age model (it might not be necessarily obvious to all CP readers). It could be reformulated such as "the NGRIP ice core δ18O record displayed on the GICC05modelext age scale".

AC: This has been updated accordingly so that the first mention of the chronology explains it clearer ....." Greenland Ice Core Chronology (GICC) 05modelext and Antarctic Ice Core Chronology (AICC) 2012" P1 L26-27

Line 21: "...too young" and "...a longer duration". By how much? Please be quantitative here and provide at least an order of magnitude.

AC: This has been revised to ...." transitions in the AICC2012 chronology occurred up to 3,000 years later than in NALPS19." P1 L28-29

- For clarity purposes, I think it is important that throughout the manuscript, the authors specify "calcite δ18O" when mentioning the δ18O records from the different speleothems and "ice δ18O" and referring to the δ18O from ice cores. They do it in places, but I think this should appear systematically to avoid any confusion.

AC: This has been updated accordingly apart from where d18O is talked about in general in both ice and speleothems.

- P2, line 40: While for further details, the reader can certainly be referred to the Erhardt et al. (2019), a few sentences need to be added to describe the added value of performing such analysis and the general principle and method used for the ramp-fitting of the transitions.

AC: These finer details are now removed from the introduction. We have, however, addressed the point as follows… "The ramp-fitting function is similar to the one used by Mudelsee (2000), but instead uses probabilistic inference to define a transition via a linear ramp between two constant levels. Such an approach enables the accurate chronological quantification of climate transitions (Mudelsee, 2000), as well as the consistent treatment of records, unlike the more subjective approach of taking the first data-point that deviates from the baseline of the previous climate state (e.g. Capron et al., 2010a; Moseley et al., 2014; Rasmussen et al., 2014), which is often not so ambiguous. „ P7, L25-30

- From P5, line 25: "Results" section: SI Table 1 could appear in the main manuscript and information could be removed from the section. I found the information provided in the text very technical and from a non speleothem expert view, I feel that this should better be the supplementary material. Instead the results section could be focused on the description of the different records and a detailed comparison of the timing inferred for the transitions in the paleoclimatic records using the statistical tool of Erhardt et al. (2019).

AC: We have removed this data and placed it in a summary table that still belongs in the main manuscript (Table 2). Additionally we have greatly expanded the discussion and provided a detailed comparison of the timing as defined by the ramp fitting. See section 5.2, P7-8

The authors discuss the relationship between calcite δ18O and calcite δ13C and perform Hendy test. Again, from a non-expert point of view, I would find it very useful to have a few of sentences explaining why they are performing such exercise, what they expect to be able to decipher from such investigation and finally what are the implications of the results of their test.

AC: This was already included in the manuscript but we have expanded with the following….

"In addition to the main isotope track along the central axis, Hendy tests (Hendy, 1971) were also prepared for each sample as a first-order assessment of whether the respective stalagmite was deposited under conditions of isotopic equilibrium, though the preferred approach in recent years has been to reproduce the data in a second stalagmite (Dorale and Liu, 2009). Under the 'Hendy test' criteria, $\delta^{18}O_{calc}$ values should remain constant along a single growth layer, and there should be no correlation between $\delta^{18}O_{calc}$ and $\delta^{13}C_{calc}$ that might otherwise indicate kinetic fractionation." P.5 L.6-11

- P8, line 13: sentence starting with "Furthermore, despite…". Is there any explanation why the St Beatus records would record a signal that is different from the Gassel samples? It would be useful to provide more information on this.

AC: Ultimately we do not know why they would be different. This would require a completely different study and extensive monitoring. We have added some explanation that is it likely due to local effects as follows….

"Furthermore, the pattern of $\delta^{18}O_{calc}$ shifts across the whole interstadial 24 to 23 period is remarkably similar in the new speleothems analysed here to the pattern of events in NGRIP $\delta^{18}O_{ice}$ across the same period. This suggests the new speleothem samples are capturing a bigger-scale climate signal, unlike EXC3 and EXC4 from St. Beatus cave (Boch et al., 2011), which show a distinctly different pattern in $\delta^{18}O_{calc}$ across this time period. The reason for the difference is unknown, and is likely due to some local influence or control at the cave site." P7, L6-11

- P9, line 5: The paragraph regarding the durations of GS-22 and the precursor event is difficult to follow. It would be very helpful if the authors could provide a table that summarises the different existing and new estimates of

the durations of GS-22, GS21.2 and GI-21.2 in the discussed paleoclimatic records. Implications from their new NALPS composite curve should be expressed more explicitly.

AC: The text has now been revised so that it is clearer (P9, L11-32) and the data is available in table 4.

- The numbering of the different sections needs to be revised. There should not be a sub Section1.1 if there is no Section 1.2 within the introduction section. I have a similar comment with the sub section 3.2.1, 3.3.1 and 3.3.4.

AC: The sub-sections are now removed.

Figure 3.c. A sentence to explain the shift in calcite δ18O values between the Asian monsoon composite records and the original data from which it was constructed need to be added.

AC: We don't think it is necessary to go into details of what other authors have done in their studies. Nevertheless, we have added "In Cheng et al., (2016), the Dongge and Hulu $\delta^{18}O$ values are reduced by 1.6 ‰ in the composite record to match the Sanbao record of Wang et al., (2008)." to the caption for figure 3..

- Figure 3. In the last sentence of the caption, "NGRIP nomenclature" should be replaced by "the latest INTIMATE event stratigraphy scheme".

AC: this has been done and the INTIMATE event stratigraphy scheme is also referred to more explicitly in the main text.

- Figure 4. This figure needs to be reworked to improve its readability. A y axis scale is missing. Transitions should be numbered following the INTIMATE event stratigraphy scheme and it should probably also show the reference curves in the background the reference curves onto which they have performed the analysis. Also, it needs to be clarified what are the three panels in (c), which speleothem records have been used to perform the transition analyses. Again, this would be straightforward if the original curves were shown underneath or in parallel.

AC: Figure 4 has been completely reworked and is now more readable. The transitions are available in the supplementary information as fig. 7 (as in the original manuscript)

- Figure 6. This figure is hard to read. Efforts must be made to improve its clarity. For instance, a triangle symbol should not be used to represent different parameters e.g. in (a), the catchment elevation relative to longitude and in the other panels some stadial δ18O values. For panels (e) and (g), "specific time periods" is vague, they should be specified. As far as I understand the caption for panel (f) is incorrect as only the mean δ18O values for the speleothems covering some selected GI and GS are being shown relative to the catchment elevation and not all. Finally, the expression "Colours are the same as in (a)" doesn't need to appear after the description of every panel. The authors could simply write the colour code at the start of the caption stating that it is the same on all the panels of the figure.

AC: With the simplification of the discussion we have been able to remove three of the sub-graphs which has greatly improved its clarity. We now have the same symbols in each graph and also a legend.

- Figure 7. The authors should be explicit on which type of δ18O values they are showing on the title of the axis e.g. ice δ18O for (a), calcite δ18O for (b) benthic δ13C and planktic δ18O.

AC: This has been added accordingly

-Figure S7. More information must be given to understand clearly what is represented: Titles for the two y axes should be provided as well as a description in the caption of the different curves that are represented e.g. δ18Odata, uncertainty ranges, probability density plots about the onset, mid-point and end of transition etc.

AC: This has been added accordingly

P1, line 16: a space is missing between "using," and "eleven ".

AC: addressed, P1 L18

P1, line 21: Since it is not mentioned previously, it is important here to specify that AICC2012 refers to an ice core chronology i.e. "NGRIP ice δ18O when displayed on the AICC2012 ice core chronology ", or alternatively, the acronym can be spelt out.

AC: addressed, P1 L26

P1, line 28: "precursor" instead of "pre-cursor".

AC: addressed, P1 L40

P1, line 29: to write "GS-24.2 COOLING event".

AC: addressed, P2 L1

P1, line 29: "...occurred shortly". Please be more specific so we have an idea from the abstract if you are talking about a few decades, or a few centuries, etc.

AC: addressed as follows…."$\delta^{18}$O depletions occurred in the decades and centuries following rapid rises in sea level" P2, L2

P1, line 35: write "orbital-".

AC: addressed, P2 L9

P2, line 2: "...have been shown to be synchronous within dating uncertainties", please provide a reference to support this statement.

AC: addressed, P2, L17

P2, line 7: There is no need for higher resolved ice δ18O profile to identify the decadaland centennial-scale variability, it was already visible from the δ18O profile published in NGRIP project members 2004. Only that no one had provided a specific description before the study by Capron et al. (2010). Hence, I think that the sentence should be rephrased.

AC: addressed as follows "In total, 25 such cycles of rapid warming and gradual cooling, as well as many other smaller centennial- and decadal-scale events, are recognised as having occurred during the last glacial period (Dansgaard et al., 1993; NGRIP Project members, 2004; Capron et al., 2010a)." P2, L23-25

P2, line 9: write "centennial-".

AC: addressed P2, L24

P2, line 14: "GICC05...." Add information regarding the time interval covered by each of the timescales.

AC: addressed, P2, L31

P2, line 32: the authors are correct about the age differences between the different chronologies and they should provide a quantitative estimate of them (at least an order of magnitude).

AC: we consider the preceding discussion to be a sufficient explanation of the differences in chronologies

P2, line 35: add a space between (ka) and (Boch et al. 2011).

AC: addressed P3, L16-17

P2, line42: "a good agreement". Please be quantitative here regarding the agreement.

AC: This section is now removed entirely

P3, line 3: why 1.1 Regional climate while there is no 1.2 and it follows the long introduction that doesn't have a sub -section heading.

AC: Regional climate is updated to its own heading 2. P3

P3, lines 17, 20 and 21: Northern Alps and Southern Alps.

AC: The northern Alps are not recognised as a regional therefore should be lowercase. The Southern Alps are recognised as a region and therefore should be capitalised.

P3, line 23: The formulation is awkward and should be rephrased with a more direct style.

AC: Addressed as follows "In particular, the phase of the North Atlantic Oscillation (NAO), which is especially pronounced in winter (Wanner et al, 1997), exhibits one of the strongest controls." P4, L4-6

P3, line 32: space is missing between (2015) and (though.

AC: addressed, P4, L15

P5, line 26: for clarity purposes, please write instead "samples from Baschg Cave" and similarly in the titles of sub-sections 3.2, 3.3, 3.4 and 3.5.

AC: this information is now in table 2

P8, line 29: here and throughout the manuscript: Erhardt et al. 2019 (not 2018).

AC: addressed, P7, L24

P9, line 2: In the paper by Columbu et al. (2017), a well-dated Sardinian speleothem covering GI-25b and GI-25a is presented. The timing of the abrupt transitions is also discussed and compared relative to the timing of the same events when displayed on the different Greenland ice core timescales. This study also provides evidences that there is a good agreement between the transition timing in the speleothem record and when considering GICC05modelext timescale, but that when considering the AICC2012 chronology, ages are younger by several millennia. The authors should mention this study in their manuscript.

AC: There are many well-dated speleothems over this time range. We do not intend to provide a review of them all and accordingly also do not pick out a select one.

P9, line 3: The sentence should be completed: "…too young by about XX yrs".

AC: We have added significantly more information as follows…" Elsewhere, the transitions in the NALPS19 chronology are consistently earlier than their counterparts in the AICC2012 chronology i.e. GS-20 (c. -400 years), GI-21.1 (c. -500 years), GI-23.1 (c. -1,900 years), GS-24.1 (c. -2,700 years), and GI-24.2 (c. -2,960 years) suggesting that some revision of the AICC2012 chronology may be needed." P8, L31-34

P9, line 7: The formulation of the sentence starting by "This demonstrated...." is awkward. It needs to be reformulated.

AC: This sentence has been removed.

P9, line 13: I don't find the information in brackets necessary, it can probably be removed.

AC: We do consider the datum to be very important, however, this has now been moved to the caption of table 4.

P9, line 22: I find the title of the section 4.3 quite vague and not really appropriate. The authors should try and be more specific.

AC: This has been updated to "5.3 NALPS δ18O variability during the last glacial period (15-120 ka)" P9, L33

P12, line 11: centennial

AC: Addressed, P11, L39

P12, line 14: space between (Fig. 7) and (Capron...).

AC: Addressed, P12, L2-3

P12, lines 13 and 27 and P13, lines 17, 18, 20 and 22: The use of the word "termination" should be avoided in this context and replaced by e.g stadial-interstadial transition. Indeed, as the authors know the word "termination" is classically used in paleoclimatology to refer to glacial-interglacial transitions and I think for clarity purposes, it is preferable to avoid introducing this term in a different context and to refer to a different climatic event.

AC: Addressed, termination has been removed in this context and replaced with "a stadial-interstadial transition"

P12, line 19: "Changes IN Ca2+" rather than "Changes TO Ca2+".

AC: Addressed, P12, L8

Also, I don't find the reference to Rasmussen et al. (2014) appropriate in this context. Instead I would suggest referring to the studies by Ruth et al. (2007). U. Ruth et al., Geophys. Res. Lett. 34, L03706, 10.1029/2006GL027876 (2007).

AC: Addressed, P12, L9-10

P12, line 25: "the NGRIP nomenclature" should be replaced by "the INTIMATE event stratigraphy scheme".

AC: Addressed, P12, L16

P13, line 13: space between (Wang et al. 2004) and (Fig. 7).

AC: Addressed, P13, L8-9

AC: We are not sure what the reviewer intends us to do as these references are all included in the manuscript

**General Comments of Reviewer 2**

My main point of critique is in the sometimes lengthy discussion, which can be difficult to follow for non-experts in both ice cores and speleothem science. I therefore suggest the authors provide moderate revisions (see specific and technical comments) to the manuscript before it can be accepted for publication

AC: We have removed two major sections and replaced them as Table 1 and Table 2

**Specific Comments of Reviewer 2**

Discussion of chronology (section 4.2): This section is in parts difficult to follow, especially for readers not overly familiar with the ice core literature. Some studies are cited, but the reasoning for this is not explained, and this can be confusing. I suggested some instances where some more background would be beneficial to improve overall clarity (see technical comments).

AC: We have added background details to the different chronologies as follows.

"For this study, ramp fitting was applied to: (1) δ18Ocalc of the new NALPS19 record (this study); (2) δ18Ocalc of the Asian monsoon composite speleothem record (Cheng et al., 2016); (3) NGRIP δ18Oice on the GICC05modelext chronology, which is comprised of a composite layer-counted chronology to 60 ka (Svensson et al., 2008) followed by splicing of the ss09sea-modelled chronology (Johnsen et al., 2001) between 60 to 122 ka onto the younger annual-layer counted chronology (Wolff et al., 2010); (4) NGRIP δ18Oice on the AICC2012 chronology, which is constructed using glaciological inputs, relative and absolute gas and ice stratigraphic markers, and Bayesian modelling (Veres et al., 2013), and; (5) NGRIP δ18Oice on the AICC2012 chronology updated by aligning δ18O of the atmosphere as measured in EPICA Dome C with δ18Ocalc of Chinese speleothems (Extier et al., 2018)."

P7, L31-40

This is especially the case for the discussion of GS-22. I think it would be worthwhile to restructure this paragraph and clarify the main message, i.e., NALPS19 allows to reevaluate conflicting results from different ice core age modelling techniques, and this is especially clear for the interval between GS-22-GS-21.2.

AC: This paragraph has been completely re-written and a table also added for clarity (Table 4). P9, L11-32

Discussion of palaeoclimate and d18O (section 4.3): I am a bit confused with the treatment of the Siebenhengste record. At the beginning of the section, the authors exclude the LGM part of the record from Siebenhengste from their discussion on the range in d18O, because of the influence from different moisture sources previously inferred for this time period. Here I was hoping the authors could provide some more background as to why this moisture source effect is only seen in the 7H LGM record: is it due to the time period covered or is the location of the cave the likely reason for this? Why are the authors certain that changes in moisture source were not an issue for any of the other records in the compilation?

AC: Siebenhengste is not excluded on the ground of moisture source but on the grounds of a different transport pathway. Full details are given in the cited manuscript as to why this occurred so there is no need to repeat it here. Furthermore, Leutscher et al, claim that the different transport pathway only affected the LGM and not other parts of the record thus our records are not affected in the time periods we are dealing with. The strong

synchronicity between Greenland and NALPS is further support that the moisture source and wider North Atlantic climate remained the same during our period of interest.

Further along in the text, there is a lengthy discussion of why the Siebenhengste record is anomalous, but there is no more mention if the source effect. I think it would greatly benefit the flow of the manuscript if the authors could elaborate a bit more on their reasoning for this, and link it back to the beginning of the section and the discussion on source changes during the LGM.

AC: The reviewer completely misses the point that the lengthy discussion is related to two caves at the same location but different elevations. Moisture source changes cannot be responsible for local variations because ultimately the moisture source would have been the same for both caves.

Discussion of stadial-level centennial-scale cold events (section 4.4): I think these events need to be more clearly pointed out in the figure 7, or even in a separate, zoomed-in figure, as it is not particularly clear what is meant now.

AC: These are already marked on figure 7 but extra text has been added to the caption accordingly "Centennial-scale cold reversals of 16.2, 17.2, 21.2, 23.2 and 24.2 are highlighted as vertical dashed yellow bars."

**Technical Comments of Reviewer 2**

Page 1: - line 21: The meaning of AICC2012 needs to be specified, otherwise this sentence is very confusing for non-experts.

AC: Addressed, P1, L26-27

- line 37: please add "oxygen" to "isotopic records" to clarify what is meant.

AC: Addressed, P2, L11

Page 2: - lines 31 and following: I think here the authors must clearly state that this chronological issue is also present in the ice cores and not only between the NALPS record and the ice cores.

AC: There is already a discussion about the chronological issues in the ice cores. P3, L3-7

Page 3: - line 2: "controlling the d18O of precipitation in this region" would be more precise.

AC: this section is now removed.

- line 20: "the northern Alps receive" (instead of receives)

AC: Addressed, P4, L1

Page 4: The description of the sites and stalagmites is a bit lengthy and confusing in parts. I wonder if it would be better for the flow of the manuscript to summarise these details in a table, and streamline the text?

AC: Addressed, Table 1

Page 5: Lines 28 and following throughout the sample description: U concentrations should be in ng/g (not ug/g) according to Table 2.

AC: Addressed, this data is now in table 2

Page 6: For the caves with only one stalagmite analysed, it would be better to merge the two headings into one.

AC: Addressed, this data is now in table 2

Page 8: Lines 27 and following: please add some context here regarding the chronologies GICC05modelext and AICC2012, otherwise it is difficult to follow for readers less familiar with ice cores.

AC: Addressed as follows:

"For this study, ramp fitting was applied to: (1) δ18Ocalc of the new NALPS19 record (this study); (2) δ18Ocalc of the Asian monsoon composite speleothem record (Cheng et al., 2016); (3) NGRIP δ18Oice on the GICC05modelext chronology, which is comprised of a composite layer-counted chronology to 60 ka (Svensson et al., 2008) followed by splicing of the ss09sea-modelled chronology (Johnsen et al., 2001) between 60 to 122 ka onto the younger annual-layer counted chronology (Wolff et al., 2010); (4) NGRIP δ18Oice on the AICC2012 chronology, which is constructed using glaciological inputs, relative and absolute gas and ice stratigraphic markers, and Bayesian modelling (Veres et al., 2013), and; (5) NGRIP δ18Oice on the AICC2012 chronology updated by aligning δ18O of the atmosphere as measured in EPICA Dome C with δ18Ocalc of Chinese speleothems (Extier et al., 2018)."

P7, L31-40

Page 9: Lines 2-4: Please provide a brief explanation of what the findings of Extier are for the readers not familiar with this study.

AC: The discussion is Extier is now extended. We have applied additional ramp fitting to Extier and include discussion on P8, L34-39

Line 5: Please add the ages of the GS-22 interval here for context.

AC: The point is that this is a discussion about the different ages for GS-22, so it does not make sense to add ages here because which ones should be chosen? The ages follow in the proceeding text and new table 4.

Also, given that this is discussed at length over the next section, I would appreciate if the authors could point out this interval (and GI-GS21.2) in Fig. 3 or 5.

AC: Addressed, the nomenclature is now on Fig. 3

Lines 6-7: "Vallelonga et al. (2012) proposed the duration of GS-22 to be 2,894±198 years and GI-21.2 - GS-21.2 to be 350 ± 19 years (together 3,244 ± 199 years, two sigma error)." This sentence reads confusingly to me: I assume the authors mean that the duration of the transition between GI-21.2 and GS-21.2 to be 350 years, while the entirety of the interval is 3244 years?

AC: The text has now been revised so that it is clearer (P9, L11-32) and the data is available in table 4.

Line 9: NGRIP-EDML should be explained.

AC: Addressed, P9, L27-28

Page 10: Line 10: "The highest and lowest δ18O values for stadials and interstadials also both come from the same caves." I find this sentence confusing: the highest and lowest in general? Which cave are these values from

AC: Addressed as follows….

"the $\delta^{18}O_{calc}$ range in mean interstadial values is 5.0 ‰ (Klaus Cramer (-7.9 ‰) and Siebenhengste (-7.9 ‰) to Kleegruben (-12.9 ‰)), whilst the range in mean $\delta^{18}O_{calc}$ stadial values is slightly larger (but comparable) at 5.4 ‰ (Siebenhengste (-9.5 ‰) to Kleegruben (-14.9 ‰)) (Fig. 7a)." P10, L23-26

Line 27: I would rephrase "mean d18O" to "mean d18O of an entire record" or similar.

AC: this will make the following discussion extremely wordy, we do not think it is necessary

Line 35: "For a given location, however, Ambach et al. (1968) argued that the altitude effect cannot be the result of a difference in condensation temperature, because the condensation level should be approximately the same." I find this sentence confusing, and would also appreciate some more details on why the condensation level is the same.

AC: As it already states in the text, the caves are in the same location therefore they receive the same rain which must have condensed at the same level. P11, L18-19

Page 13: Line 32-35: "Furthermore, we suggest that the highly-debated GS-22 - GI21.2 - GS-21.2 interval had a duration of 3,857 ± 249 years, which is in closer agreement with the 4,121 ± 325 years of NGRIP-EDML (Capron et al., 2010b) and the3,793 ±805 years of the Asian monsoon composite (Kelly et al.,2006; 35 Kelly, 2010; Cheng et al., 2016)." Closer agreement than what?

AC: Addressed as follows...

Additionally, we propose that the duration of the highly-debated GS-22 - GI-21.2 - GS-21.2 interval was 3,993 ± 155 years, which is in closer agreement **to the duration of** 4,122 ± 650 years in NGRIP-EDML (Capron et al., 2010b) and the 4,489 ± 960 years of the Asian monsoon composite record (Kelly et al., 2006; Kelly, 2010; Cheng et al., 2016).

P13, L 32-35

Figure 3: In the caption, I believe there is some information missing. For c) only the Dongge data is referenced, and there is no mention of Hulu. There is also a repetition at f)"for(e)colour-coded the same".

AC: Addressed. Details have been added to say that the revised Hulu record comes from Cheng et al., 2016. The repetition has been removed.

Figure 4: I think this figure would benefit from some additional work. For example, it would be clearer if the different ice cores (b) and stalagmites (c) for which the ramp-fitting was done were indicated in the figure with labels. Also, possibly adding labels for the transitions identified in the Greenland records would help.

AC: This figure has been completely revised.

Figure 5: Here it would be helpful to the reader if the records were labelled, as in figure 2.

AC: If the records were labelled as in figure 2 it would be too busy therefore we just stick with the studies rather than individual speleothems.

Figure 6: I think it would be helpful to have a legend in the figure showing which symbol belongs to which cave.

AC: This has been added and is now fig. 7.

Yours faithfully,

Gina Moseley, Email. gina.moseley@uibk.ac.at

[revised manuscript text omitted]
 $\delta^{18}O_{ice}$ record on GICC05$_{modelext}$ chronology (Wolff et al, 2010); (d) NGRIP $\delta^{18}O_{ice}$
record on AICC2012 chronology (Veres et al., 2013); NGRIP $\delta^{18}O_{ice}$ record on the Extier et al (2018) revised
AICC2012 chronology. Each ramp-fit relative to its reference curve is given in SI Fig. 7. The GICC05$_{modelext}$
chronology does not contain uncertainties in this time period (Wolff et al., 2010) thus these errors are based on the

¶
¶

maximum counting error of Svensson et al. (2008). Extier et al (2018) quote an uncertainty of 2,440 years (2 sigma) in MIS 5. Uncertainties are not given outside of MIS5.

[revised manuscript text omitted]

---

## Author Response (AR2)

University of Innsbruck
Institute of Geology
Innrain 52
Innsbruck 6020
Austria

20th November, 2019

Dear Dominik,

I would like to submit the paper "NALPS19: Sub-orbital scale climate variability recorded in Northern Alpine speleothems during the last glacial period" with the final corrections to Climate of the Past. We would like to thank the reviewers who provided especially constructive comments throughout the review process.

**Comments of Reviewer 1**

-Page 10, line 27. With the current presentation, I don't think that the link made between ice δ18O from Greenland and calcite δ18O is appropriate. While both are likely controlled (at least partly) by temperature, this not the same "temperature" that is considered. For Greenland, Johnsen et al. (2001) refer to local temperature (i.e. at the Greenland drilling site), which is different from the temperature controlling precipitation δ18O in the northern and central Alps. The authors have to revise their statement and be more cautious and specific.
AC: agreed this has been revised and the reference to Greenland removed.

-Page 13, line 17: This sentence should be removed as I do not think the comparison of these short events with glacial terminations is appropriate. Glacial terminations are primarily forced by orbital forcing, hence the climate background conditions are fundamentally different from the short-lived events and the time-scales are also obviously completely different.
AC: we do not agree that the sentence should be removed because it is an important summary statement. We have changed 'termination' to 'stadial-interstadial transition' so that the correct terminology is used.

-Figure 4. I find Figure 4 extremely difficult to read. I appreciate that the authors modified the previous version unfortunately, I find this one even harder to interpret and currently of no added values considering the presence of Table 3. A clear caption explaining the different symbols is missing and the authors have to think of a better and clearer representation of those transitions (and consider using different colors?). As said and suggested in my first review a y axis scale is missing (or at least an explanation for what is done), and it should probably also show the reference curves in the background the reference curves onto which they have performed the analysis. Uncertainties are overlapping quite largely for the transitions displayed on GICC05modelext, AICC2012 and AICC2012 (extier) which prevent a clear reading of the figure, a different representation should hence be used. Also, it is unclear what is the error bar presented: is it the error bar (5% -95% confidence interval) provided through the rampfitting analysis? Is it the error bar associated to the chronologies? Or are the authors providing an error based on the combination of the two sources of uncertainties I mention below? At the moment, it is very unclear from what is written in the caption of the figure.
AC: What this figure does is provide an easy and simple visual representation of the results of the ramp fitting, something that table 3 does not. It is therefore important to keep it. The different symbols have now been added to the caption as well as a clear statement that they relate to the start, middle and end of the transitions as defined by the ramp-fitting. This information has also been added to the y-axis. The transitions are also separated by colours though each transition is anyhow clearly labelled across the top. The caption already refers the reader to the reference curves "Each ramp-fit relative to its reference curve is given in SI Fig. 7". These are too many and numerous and superfluous information, thus they are in the supplementary information. Caps have been removed from most error bars so that they do not physically touch, but are kept for the NALPS section because they are so small and would otherwise be lost at this scale. An explanation of the error bars is also added to the caption.

-Figure 5. I have a similar comment for Figure 5. The authors need to provide a caption right next to the graph, so it is easier to figure out what the different symbols are representing, as well as the lines between the symbols. The authors should also explain the difference between panel b and panel c?
As for panel d, it is very unclear and hard to read. I don't find it of added values considering that the results are provided in the Table 3, I would suggest removing it.
AC: Similarly this is an important visualisation of the data presented in Table 3 enabling a quick assessment of offsets and durations that is not possible with table 3. As requested a legend is now provided with the graph. An explanation of the lines has been added. Panel c is at a different scale to panel b. We consider this obvious and therefore it does not need to be stated in the caption. The durations of transitions between stadials and

interstadial and vice versa is a very important topic, thus this visual representation of this data is important and should stay.

-In general, the resolution of the figures is of quite poor quality (at least looking at the pdf provided). High-resolution figures have to be provided for the published version of the paper.
AC: This is a function of the submission process

Minor comments:
-Page 1, line 22: "…with the exception of Stadial-20". I would suggest to simply write the following: "Ramp-fitting analysis further reveals that, except for one abrupt change, the timing of d18O transitions occurred…". This will also avoid introduce a new notation such as "stadial-20".
AC: done

-Page 1, line 24: To remove "large" since the scale of the uncertainties are already quantified with "millennial-scale".
AC: done

-Page 1, line 31: To be a bit more specific regarding the "ramp-fitting" as this might not be obvious to all readers what you mean e.g. "using a ramp-fitting analysis to objectively identify the onset and the end of the abrupt transitions, we show that…".
AC: done

-Page 1, line 40: The acronym GS has to be defined.
AC: done

-Page 3, line 4: To write "In the vicinity OF …".
AC: done

-Page 6, lines 21-22: To write "six" instead of 6 and "three" instead of 3 in the next line.
AC: done

-Page 7, lines 27-29: To replace "accurate" by "objective" and to remove "which is often no so ambiguous".
AC: replacing accurate with objective would be grammatically incorrect given the remainder of the sentence. The sentence has instead been revised to
*Such an approach enables a consistent approach to chronological quantification of climate transitions (Mudelsee, 2000), unlike the more subjective approach of taking the first data-point that deviates from the baseline of the previous climate state (e.g. Capron et al., 2010a; Moseley et al., 2014; Rasmussen et al., 2014).*

-Page 8, line 41: Add a space between "the" and "AICC2012".
AC: done

-Page 9, line 7: "Interestingly, with the exception of GI-23.1, the duration of transitions in the Asian monsoon are considerably longer (on multi-centennial timescales) than for the North Atlantic–sourced NALPS19 and Greenland chronologies (on multi-decadal timescales) (Table3, Fig. 5)." Could it partly be due to a lower resolution of the speleothem records? The authors should address this.
AC: there are multiple data points within the transitions of the Asian monsoon records. It is not a product of resolution therefore we do not consider it to need particular attention.

-Page 9, line 31: The formulation "a longer-duration GS-22 - GI-21.2 - GS-21.2 period" is very awkward, please reformulate.
AC: reformulated to *"The speleothem $\delta^{18}O_{calc}$ records from both the Alps and China therefore support a longer-duration for the period between the cooling into GS-22 to the warming into GI-21.1e, which is in line with the NGRIP-EDML chronology (Capron et al., 2010b; Vallelonga et al., 2012)."*

-Page 10, line 4: Replace "an" by "a".
AC: this would be incorrect grammer because the use of a and an is based on the sound of the first letter, not just the first letter itself. Therefore MIS sounds em-ey-ess, which means it begins with a vowel sounds and should therefore be 'an MIS'

-Page 10, line 18: The sentence does not seem to be finished and there is a missing reference at the end.
AC: done

-Page 10, line 41: "as IT would be expected…".
AC: disagree, incorrect grammatically

5    -Page 12, line 30: "Such cold reversals ARE thought…"
AC: done

-Page 13, line 1: Add a space between (2012) and (Fig. 8).
AC: done
10
-Page 13, line 7: "δ18O planktic"
AC: done but as planktic δ18O

-Page 12, line 35: I believe this is the first time the authors refer to the H-events, they should be briefly defined.
15   AC: we consider this unnecessary in a journal about Climate of the Past. Readers should already be familiar with Heinrich events. The reviewer does not request definitions for other events or components of the climate system such as AMOC, Preboreal Oscillation, Yougner Dryas etc.

-Figure 2.c. Please do not introduce a new notation for stadial and interstadial events. Quite a few are circulating
20   in the litterature and this is quite confusing already. Hence, please remove the I-X and S-X. I do not think the numbering of events is necessary in this context here anyway.
AC: done

-Table 3. Write "…for NALPS δ18O (this study), NGRIP δ18O on …."
25   AC: done

-Table 4. Remove "various" or "different" to the caption, but don't keep both.
AC: done

Yours faithfully,

G E Moseley

Gina Moseley, Email. gina.moseley@uibk.ac.at

**NALPS19: Sub-orbital scale climate variability recorded in Northern Alpine speleothems during the last glacial period**

Gina E. Moseley[1], Christoph Spötl[1], Susanne Brandstätter[1], Tobias Erhardt[2], Marc Luetscher[1,4], R. Lawrence Edwards[3]

[1]Institute of Geology, University of Innsbruck, Innrain 52, 6020 Innsbruck, Austria

[2]Climate and Environmental Physics and Oeschger Center for Climate Change Research, University of Bern, Sidlerstrasse 5, 3012 Bern, Switzerland

[3]School of Earth Sciences, University of Minnesota, John T. Tate Hall, Room 150, 116 Church Street SE, Minneapolis, MN 55455-0149, USA

[4]Swiss Institute for Speleology and Karst Studies (SISKA), 2301 La Chaux-de-Fonds, Switzerland

*Correspondence to:* Gina E. Moseley (gina.moseley@uibk.ac.at)

**Abstract.** Sub-orbital-scale climate variability of the last glacial period provides important insights into the rates that the climate can change state, the mechanisms that drive such changes, and the leads, lags and synchronicity occurring across different climate zones. Such short-term climate variability has previously been investigated using $\delta^{18}O$ from speleothems ($\delta^{18}O_{calc}$) that grew along the northern rim of the Alps (NALPS), enabling direct chronological comparisons with $\delta^{18}O$ records from Greenland ice cores ($\delta^{18}O_{ice}$). In this study, we present NALPS19, which includes a revision of the last glacial NALPS $\delta^{18}O_{calc}$ chronology over the interval 118.3 to 63.7 ka using eleven, newly-available, clean, precisely-dated stalagmites from five caves. Using only the most reliable and precisely dated records, this period is now 90 % complete and is comprised of 16 stalagmites from seven caves. Where speleothems grew synchronously, the timing of major transitional events in $\delta^{18}O_{calc}$ between stadials and interstadials (and vice versa) are all in agreement on multi-decadal timescales. Ramp-fitting analysis further reveals that, except for one abrupt change, the timing of $\delta^{18}O$ transitions occurred synchronously within centennial-scale dating uncertainties between the NALPS19 $\delta^{18}O_{calc}$ record and the Asian Monsoon composite speleothem $\delta^{18}O_{calc}$ record. Due to the millennial-scale uncertainties in the ice-core chronologies, a comprehensive comparison with the NALPS19 chronology is difficult. Generally, however, we find that the absolute timing of transitions in the Greenland Ice Core Chronology (GICC) 05$_{modelext}$ and Antarctic Ice Core Chronology (AICC) 2012 are in agreement on centennial-scales. The exception to this is during the interval 100 to 115 ka, where transitions in the AICC2012 chronology occurred up to 3,000 years later than in NALPS19. In such instances, the transitions in the revised AICC2012 chronology of Extier et al. (2018) are in agreement with NALPS19 on centennial scales, supporting the hypothesis that AICC2012 appears to be considerably too young between 100 to 115 ka. Using a ramp-fitting function to objectively identify the onset and the end of abrupt transitions, we show 
[revised manuscript text omitted]
, whereas the uncertainty bars related to the original chronologies.** (a) NALPS19 $\delta^{18}O_{calc}$ record (closed circles, this study); (b) Asian monsoon composite speleothem $\delta^{18}O_{calc}$ record (open circles, Kelly et al., 2006; Kelly, 2010; Cheng et al., 2016); (c) NGRIP $\delta^{18}O_{ice}$ record on GICC05$_{modelext}$ chronology (closed upward triangles, Wolff et al, 2010); (d) NGRIP $\delta^{18}O_{ice}$ record on AICC2012 chronology (open down triangles, Veres et al., 2013); NGRIP $\delta^{18}O_{ice}$ record on the Extier et al (2018) revised AICC2012 chronology (closed down triangles). Each ramp-fit relative to its reference curve is given in SI Fig. 7. The GICC05$_{modelext}$ chronology does not contain uncertainties in this time period (Wolff et al., 2010) thus these errors are based on the maximum counting error of Svensson et al. (2008). Extier et al (2018) quote an uncertainty of 2,440 years (2 sigma) in MIS 5. Uncertainties are not given outside of MIS5.

[revised manuscript text omitted]